



# Subglacial sediment transport upstream of a basal channel in the ice shelf of Support Force Glacier (West Antarctica), identified by reflection seismics.

Coen Hofstede[1], Sebastian Beyer[1], Hugh Corr[3], Olaf Eisen[1, 2], Tore Hattermann[4], Veit Helm[1], Niklas Neckel[1], Emma C. Smith[1], Daniel Steinhage[1], Ole Zeising[1], and Angelika Humbert [1, 2]

[1]Alfred Wegener Institute, Helmholtz Centre for Polar and Marine Research, Am Handelshafen 12, 27570, Bremerhaven, Germany
[2]University of Bremen, Klagenfurter Straße 28359, Bremen, Germany
[3]British Antarctic Survey, National Environmental Research Council, Cambridge, CB3 0ET, UK
[4]Norwegian Polar Institute, Framsenteret, Hjalmar Johansens gate 14, 9296 Tromsø, Norway

**Correspondence:** Coen Hofstede (coen.hofstede@awi.de)

**Abstract.** Flow stripes on the surface of an ice shelf indicate the presence of large channels at the base. Modelling studies have shown that where these surface expressions intersect the groundling line, they coincide with the likely outflow of subglacial water. An understanding of the initiation and the ice–ocean evolution of the basal channels is required to understand the present behaviour and future dynamics of ice sheets and ice shelves. Here, we present focused active seismic and radar surveys of a

basal channel and its upstream continuation on Support Force Glacier which feeds into the Filchner Ice Shelf, West Antarctica. We map the structure of the basal channel at the ice base in the grounded and floating part and identify the subglacial material within the grounded part of the channel and also along the seafloor. Several kilometers upstream of the grounding line we identify a landform, consisting at least in part of sediments, that forms the channel at the ice base.Immediately seaward of the grounding line, the seismic profiles show a 200 m thick partly disturbed, stratified sediment sequence at the seafloor, which

we interpret as grounding line deposits. We conclude that the landform hosts the subglacial transport of sediments entering Support Force Glacier at the eastern side of the basal channel. In contrast to the standard perception of a rapid change in ice shelf thickness just downstream of the grounding line, we find a very flat topography of the ice shelf base with an almost constant ice thickness gradient along-flow, indicating only little basal melting, but an initial widening of the basal channel, which we ascribe to melting along its flanks. Our findings provide a detailed view of a more complex interaction of grounded

landforms, ice stream shear margins and subglacial hydrology to form basal channels in ice shelves.



## 1   Introduction

Ice shelf channels (Drews, 2015), also known as channels (Alley et al., 2016), surface channels (Marsh et al., 2016) or M-channels (Jeofry et al., 2018) are narrow long channels on the surfaces of ice shelves. They are often remotely detected as flow stripes with satellite imagery like MODIS, (Moderate Resolution Imaging Spectroradiometer, (Scambos et al., 2007)) or

Landsat 8. These channels are a surface expression of a sub-ice shelf channel (Le Brocq et al., 2013), also known as basal channel (Marsh et al., 2016; Alley et al., 2016, 2019) or U-channel (Jeofry et al., 2018), most often aligned to the ice flow direction but occasionally migrating across the ice flow direction. As locations of thinner ice these channels can induce ice shelf fracturing (Dow et al., 2018). Thus ice shelf channels potentially influence ice shelf stability, which in turn provides stability of the ice sheet through the buttressing effect (Thomas and MacAyeal, 1982; Fürst et al., 2016). Alley et al. (2016)

categorized three types: (1) ocean sourced channels that do not intersect with the grounding line, (2) subglacially sourced channel that intersect with the grounding line and coincide with modeled subglacial water drainage and (3) grounding line sourced types that intersect the grounding line but do not coincide with subglacial drainage of grounded ice. (In the following we will use the term surface channel and basal channel to make a clear separation.)

In the grounding line area of the Antarctic Ice Sheet, the location of modeled channelized meltwater flow at the Antarctic ice

sheet often coincides with basal channels (and its surface expression, the surface channel) in the grounding line area (Le Brocq et al., 2013). This suggests subglacial drainage contributes to the formation of sub-ice channels. According to Jenkins (2011) subglacial meltwater entering the ocean cavity at the grounding line forms a plume entraining warmer ocean water and causes increased subglacial melt beneath the ice shelf which drives the further evolution of channel geometry. This hypothesis is often graphically supported by an idealized and conventional geometry of the sheet–shelf transition at the grounding line area: the

ice–water contact (the underside of the ice shelf) rises steeply passing the grounding line, thus allowing fresh water influx to form uprising melt plumes and then leveling out more horizontally further downstream (Le Brocq et al., 2013; Drews et al., 2017).

Drews et al. (2017) linked the formation of a basal channel to a potential esker upstream of the grounding line, noting that the channel dimensions are an order of magnitude larger then eskers in deglaciated areas. However, Beaud et al. (2018) found

that eskers are more likely to form under land terminating glaciers. Jeofry et al. (2018) concluded that the basal channels at Foundation Ice Stream were formed by hard rock landforms upstream of the grounding line. Bathymetric surveys at different locations showing hard bedded landforms of similar dimensions as the basal channels confirmed this as a possibility. Alley et al. (2019), however, argued that shear margins of ice streams develop surface troughs continuing downstream of the grounding line. Once afloat, these surface troughs become spots at the ice shelf base when adjusting the hydrostatic equilibrium, thereby

forming a basal channel. Thus a channelized warm water plume is likely to incise a basal channel forming observed polynyas at the ice shelf front. Both hard rock bedforms and surface troughs at shear margins of ice streams seem to cause basal channels. Unfortunately, often key observations are missing, e. g. on the type of material and structure of the bed upstream of a basal channel.





From noble gas samples at six locations beneath the Filchner Ice Shelf, Huhn et al. (2018) estimated an total freshwater influx of 177 ± 95Gt/a, entering the Filchner Ice Shelf. At one location, downstream of Support Force Glacier (SFG), the noble gas sample indicated crustal origin and thus part of the freshwater influx having a grounded subglacial origin. We also know that the west side of SFG, where an surface channel is present, coincides with modeled channelized subglacial drainage

(Le Brocq et al., 2013; Humbert et al., 2018). Thus we have good reason to assume there is subglacial drainage present at SFG.

Most field observations of surface channels and basal channels come from satellite imagery, airborne or ground penetrating radar. Although airborne radar gives a good impression of the shape of the ice shelf at larger scales, its trace distance is large (10 m) and primarily registers nadir reflections. The narrow aperture thus provides only limited insight in the precise geometry of the channel, especially steep structures like the flanks of the basal channel. In addition radar signals do not penetrate below

wet ice-bed contacts, making it hard to determine the nature of subglacial material: is water exclusively present on hard bedrock or does the substrate also consist of sediments in which case we can expect recent sedimentation on the seabed?

To investigate the ice–bed characteristics we deployed an active, high-resolution seismic survey on the west side of the sheet–shelf transition of SFG (Fig. 1). The highly resolved seismic signal allows to reconstruct complex subglacial structures like basal channels properly, given the larger aperture of the system compared to airborne radar. It also informs us about subglacial

and ocean floor properties, as the signal penetrates through water containing substrata and the sub-shelf ocean cavity. This seismic survey is backed up by airborne radar data of the sheet–shelf transition of SFG collected in 2017. Key questions we want to answer are: What initializes the surface and basal channel? How does it proceed upstream of the grounding line? Is there any subglacial drainage connected to the channel?

We first discuss the survey site and the different data sets, then the results of the seismic data analysis. Finally, we discuss

the possible interpretations of our findings, including subglacial landforms.

## 2   Survey area and data

### 2.1   Site description

SFG is an ice stream in West Antarctica feeding into the Filchner Ice Shelf. The ice stream lies between Foundation Ice Stream in the southwest and Recovery Glacier in the northwest (Fig. 1). The northward ice flow is tugged in between the Pensacola

Mountains on the western side and the Argentina Range on the eastern side constraining the ice shelf for 50 km. The drainage basin of SFG is not that well defined (Rignot et al., 2011). Although it drains from interior East Antarctica, it is linked to West Antarctica through the Filchner-Ronne Ice Shelf (Bingham et al., 2007). At the grounding line (GL) the ice is grounded 1200-1400 m below sea level (mbsl, WGS84 ellipsoid) with a surface velocity of 200 m/a. Upstream of the GL the bed is retrograde, it dips gently (slope of 0.28°) for some 20 km followed by a 400 m rise over the next 10 km and a fairly constant

depth over the next 30 km (Fretwell et al., 2013). The survey target was an isolated surface channel (at the surface of the ice shelf) and its basal channel (at the base of the ice shelf) on the western side of SFG, not influenced by other basal channels which might affect the ice dynamics or ice–seawater interaction. At the GL we performed a high-resolution seismic reflection survey consisting of two along- and three across-profiles forming a grid.





## 2.2 Surface velocities and grounding line position

Surface velocities were combined from Landsat-8 and TerraSAR-X derived velocity fields. Landsat-8 velocity fields were downloaded in near real time (Scambos et al., 2016). Here preference was given to 64 day repeat passes as a trade-off between accuracy and decorrelation (Fahnestock et al., 2016). Due to orbital constrains Landsat velocity estimates reach a maximum

latitude of ∼82.7°S which is just upstream the grounding line of SFG (Figure 1a). In order to extend the velocities further south, we employed additional data takes from TerraSAR-X acquired in left looking mode. TerraSAR-X surface velocity fields were calculated by means of intensity offset tracking on single look complex imagery (e.g. Strozzi et al., 2002). Subsequently all velocity fields were filtered by the three step filtering procedure introduced by Lüttig et al. (2017) and merged into a continuous velocity mosaic. Employing the same TerraSAR-X data as in the calculation of the velocity fields we were able

to generate several coherent double differential interferograms which were used to slightly modify grounding line locations (Fig. 1c) obtained from DLR (e.g. Rignot et al., 2011).

## 2.3 Airborne radar data

In January 2017, the British Antarctic Survey (BAS) collected airborne ice-penetrating radar data with the PASIN2 system (an upgraded version of that described by Rippin et al. (2014). The radar acquired data with a repetition frequency 3125 Hz, which

was then pre-stacked and processed with an unfocused SAR algorithm before being decimated to an equivalent along-track spacing of 11 m. The onset of the bed reflector was then obtained with a semi-automated process and merged with a laser terrain mapper to give the ice thickness and bed elevation: a wave speed in ice of 0.168 m $\mu s^{-1}$ along with a 10 m firn correction was used.

## 2.4 Routing of subglacial water

To determine the subglacial water pathways we used a simple flux routing scheme to compute subglacial water pathways as described in Humbert et al. (2018). Subglacial water flow is governed by the hydraulic potential Φ, which can be written as

$$\Phi = \rho_w g h_b + \rho_i g H, \tag{1}$$

where $\rho_w$ is the density of water, $g$ acceleration due to gravity, $h_b$ bed elevation, $\rho_i$ density of ice and $H$ the ice thickness.

For the bed elevation $h_b$ we used a combination from BEDMAP2 (Fretwell et al., 2013) and the airborne radar data. The

airborne radar profiles were nested into the BEDMAP2 dataset using the continuous curvature splines in tension algorithm implemented in the Generic Mapping Tools (GMT, Smith and Wessel, 1990). Next to our airborne radar data we incorporated all regionally available Operation IceBridge MCoRDS L2 ice thickness measurements in our analysis (Paden et al., 2019). In order to achieve a smooth transition between BEDMAP2 and the radar data we further included data points from the gridded BEDMAP2 dataset within a 50 km buffer in the interpolation.

The modeled water routing (Fig. 1b) shows expected subglacial drainage routes entering the ocean cavity of SFG. In agreement with Alley et al. (2019), three influx entrances are predicted at the shear margins of SFG: two smaller ones on the eastern



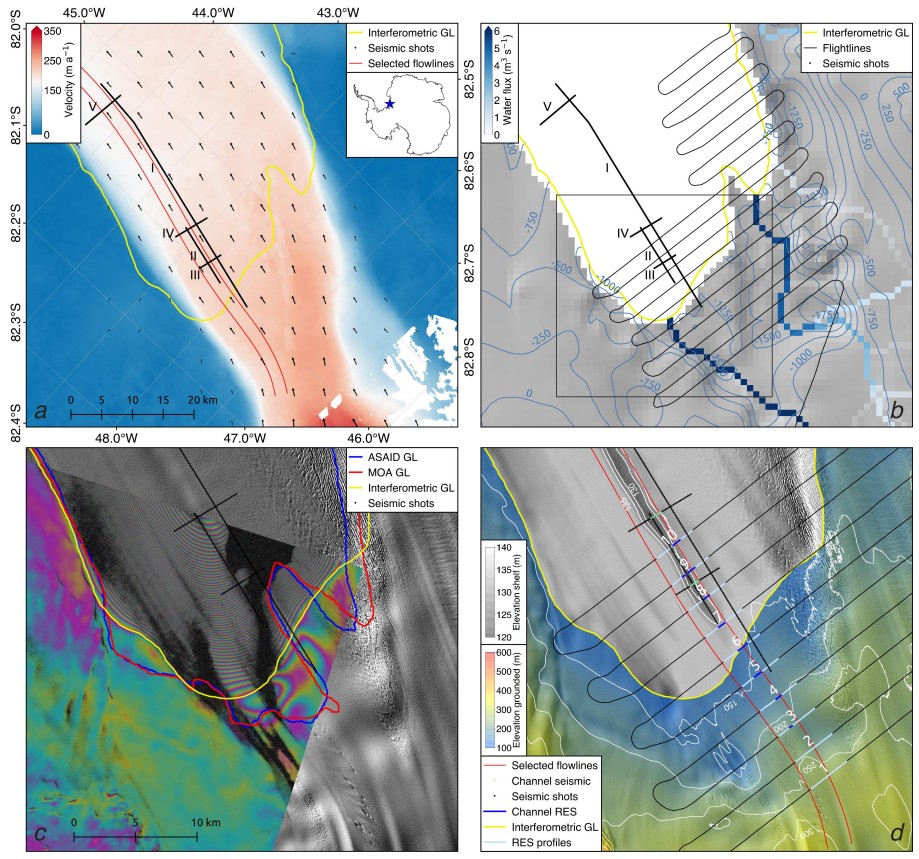

**Figure 1.** Location of seismic and airborne radar survey of the GL area of SFG. (a) Surface ice velocity map of the survey area. The seismic survey grid is marked by black dots, along-profiles I and II and across-profiles III, IV and V. Two flow lines are marked in red and the GL in yellow. Inset: location of SFG in Antarctica. (b) Modeled subglacial water routing flux at SFG from the static hydrological potential. Shown are our updated bedrock compilation, a combination of BEDMAP2 and the collected airborne radar survey (thin black looping line). The black rectangle marks the subregions of (b) and (c). (c) Three proposed GLs are marked in blue (ASAID Bindschadler et al. 2011), red (MODIS Scambos et al. 2007) and yellow (based on interferometry). The background is a shaded version of the 8 m Reference Elevation Model of Antarctica (REMA, Howat et al. 2019) overlaid by a TerraSAR-X interferogram used in the delineation of the grounding line. (d) The topography of the ice shelf (grey) indicates the surface channel. Numbered are loops 1 to 10 of the airborne radar data. Loops 1 to 5 are on the grounded part. Loops 7 to 10 represent the shelf, loop 6 is at the GL. In light blue we see the radar across-profiles shown in Figure 6. In green (seismic profiles) and dark blue (radar) we see the shot locations at the sub-ice channel.

side and one larger influx on the western side, close to the surface channel and the seismic survey area. Our seismic survey focussed on the larger influx entrance on the western side of SFG.



**Table 1.** Properties of the five collected seismic profiles of SFG

| Profile | length (km) | Source | Direction | Position from GL |
|---|---|---|---|---|
| I | 43.5 | 10 m (100 g) detonating cord | along-flow | 3.90 km upstream of GL |
| II | 10.0 | 10 m (100 g) detonating cord | along-flow | 3.50 km downstream of GL |
| III | 4.2 | 150 g cartridge in borehole | across-flow | 7.61 km downstream of GL |
| IV | 6.0 | 10 m (100 g) detonating cord | across-flow | 14.15 km downstream of GL |
| V | 7.5 | 10 m (100 g) detonating cord | across-flow | 38.45 km downstream of GL |

## 2.5 Seismic data recording

In total we collected 71 km of seismic data divided over five profiles, numbered I to V, forming a grid (Fig. 1a, b). We used a 300 m snow streamer with 96 gimballed 30 Hz vertical p-wave sensors, pulled behind a Nansen sledge carrying the recording equipment (four Geodes recording 24 channels each). The record length was set to 5 s and the sample rate to 0.5 ms. We used

a snowmobile to pull the sledge and streamer between shot locations. As a source we mostly used 10 m detonating cord of 10 g/m (so each shot used 100g PETN) placed 34 m in front of the near offset geophone and parallel to the snow streamer. At across-profile II we used 5 m deep drilled boreholes filled with 150 g pentolite cartridges. The shot spacing was half a streamer length, 150 m, resulting in single-fold data coverage. We refer to the single-fold data as profiles and collected two along-profiles and three across-profiles (Table 1).

During the data acquisition we collected 13 long offset gathers. At these shot locations we placed four shots of detonating cord at one location and recorded the shots at continuously decreasing offsets:

- Shot 1: offset 934 to 1234 m, streamer 934 m from the shot

- Shot 2: offset 634 to 934 m, streamer 634 m from the shot

- Shot 3: offset 334 to 634 m, streamer 334 m from the shot

- Shot 4: offset 34 to 334 m, streamer 34 m from the shot, profiling configuration

We used Shot 4 both in the long offset gather and for profiling. We created three types of processed data sets for the following applications:

- Profiles: Here the processing aim is to get an $xz$-image ($x$-axis: horizontal and $z$-axis: vertical) revealing the dimension and structure of the ice, ocean cavity and seabed. The data were band pass filtered (30–540 Hz), stacked, Kirchhoff time

migrated and depth converted. Especially depth conversion of the time migrated profiles is important as the sub-shelf channel area has considerable topography over a relative flat seabed. As the seawater is a slow p-wave velocity layer, thickness variation in the water column induces time delays in the underlying seabed and an apparent seabed topography in the time migrated profiles.





- Single shot gathers to determine the seismic reflection coefficient $R$: Here the aim is to map the amplitude values of different subsurface reflections and of the direct wave of raw shot gathers. Except for adding a geometry, these shots were not processed as any processing affects the amplitudes.

- Long-offset gathers: Here we combine four shots with sequentially increasing offset into one shot location with a long offset. The aim here is to register the Normal Move Out (NMO) of a reflection for long offsets (time delay of a reflection with increasing offset) from which subsurface seismic p-wave velocities can be derived. The data were processed such that the reflections are best visible. Processing steps include muting, spiking deconvolution, band pass and fk-filtering.

## 2.6 Seismic reflection and transmission coefficient at normal incidence:

Reflectivity at a planar and specular two media interface in the subsurface depends on contrast of P-wave velocity ($V_p$), S-wave velocity ($V_s$), density ($\rho$) and the angle of incidence ($\theta$) at the interface of the two considered media:

$$R(\theta) = \frac{A_1(\theta)}{D A_0} r(\theta) e^{\alpha r(\theta)}, \tag{2}$$

with $A_1(\theta)$ being the amplitude of the primary reflection and $A_0$ being the source amplitude, $D$ a directivity factor caused by the use of detonating cord as a source (when using point sources such as borehole shots $D = 1$), $r(\theta)$ the distance of the primary wave and $\alpha$ the seismic attenuation coefficient. These quantities can all be determined from single shot records. The directivity factor $D$ is discussed below in the determination of $A_0$ subsection.

With a target depth of 1400 m or deeper, and an offset ranging of 33 to 330 m, we get $0.6° \leq \theta \leq 6.7°$. Thus we approximate the considered reflections of the profiling shots are normal incidence. A shortcoming of using a relative small spread is that we are not able to plot the $\theta$-dependency of subglacial or seabed materials making identification less certain.

At normal incidence the reflection coefficient is solely determined by the contrast of the acoustic impedance ($Z = \rho V_p$) at the media interface:

$$R = \frac{Z_2 - Z_1}{Z_2 + Z_1} = \frac{\rho_2 V_{p_2} - \rho_1 V_{p_1}}{\rho_2 V_{p_2} + \rho_1 V_{p_1}}, \tag{3}$$

where subscripts 1 and 2 refer to upper and lower media. In Table 2 we consider $R$ at normal incidence from the following media interfaces we encounter in our survey area:

- grounded ice–bed interface;

- shelf ice–seawater interface;

- seawater–seabed interface;

In all cases of considered media interfaces, the acoustic impedance of the upper medium, ice (grounded or shelf ice) or the sub-shelf seawater can be estimated quite accurately as the material (ice or seawater) is known and the acoustic impedance of the lower medium (subglacial material or seabed) is unknown. Using both equations we can determine the acoustic impedance of subglacial material and the seabed of single shots.



To calculate $R$ at the seawater–seabed interface ($R_{s-b}$, where subscripts $i, s$ and $b$ refer respectively to *ice*, *seawater* and *bed* (both the bed upstream of the GL and seabed), respectively) we assume normal incidence. The smallest possible value for $R$ is caused by an ice–seawater transition; with $Z_i = 3.44 \times 10^6$ kg/m²s and $Z_s = 1.45 \times 10^6$ kg/m²s we get:

$$R_{i-s} = \frac{Z_s - Z_i}{Z_s + Z_i} = \frac{(1.45 - 3.44) \times 10^6}{(1.45 + 3.44) \times 10^6} = -0.41. \tag{4}$$

The transmission coefficient $T$ is given by:

$$T = \frac{2Z_1}{Z_2 + Z_1} = \frac{2\rho_1 V_{p_1}}{\rho_2 V_{p_2} + \rho_1 V_{p_1}}. \tag{5}$$

To calculate $R_{s-b}$ we must take into account the energy loss at the ice shelf–seawater interface. To compensate for this energy loss we assume normal incidence and an abrupt transition at the ice–seawater interface. Under these assumptions and with the ice–seawater transmission coefficient ($T_{i-s} = 1.41$) and seawater–ice transmission coefficient ($T_{s-i} = 0.59$) we get

$$R_{s-b} = \frac{A_1(\theta)}{A_0 T_{i-s} T_{s-i}} r(\theta) e^{\alpha r(\theta)} = \frac{A_1(\theta)}{0.83 A_0} r(\theta) e^{\alpha r(\theta)}, \tag{6}$$

with $A_1(\theta)$ being the amplitude of the seabed reflection.

### 2.7  Seismic attenuation of the ice and seawater

To determine the seismic attenuation $\alpha$ we need an estimate of the temperature of the shelf ice. We used temperature data from the 862 m long borehole FSW2 at the Filchner Ice Shelf at $-44.22546°$W, $-80.56532°$S, about 190 km downstream (northwest) of our survey area and another 275 km from the calving front of the Filchner Ice Shelf. The installed thermistor chain showed an ice temperature range between –29°C at 10 m depth to –24°C at 650 m depth then increasing to –2.5°C at the base. As the ice shelf at the survey area is thicker (1300 m), we extrapolated the temperature curve giving and average seismic
attenuation of 0.2/km for the entire ice column.

We used the seawater temperature from the same borehole data, -2.3°C, to calculate the seismic attenuation of the water column. Assuming a constant temperature for the entire subglacial seawater column, we get an attenuation of 0.001 dB/km (Ainslie and McColm, 1998). This converts to $1.15 \times 10^{-4}$/km which is so low that we can ignore this component.

Values of $R$ for media contrasts we most likely encounter are listed in Table 2. In general, one can say that the higher the
water content of the lower medium, the smaller is $R$. If ice is the upper medium, the range of $R$ is larger than if the seawater is the upper medium, making the interpretation of the seabed more sensitive to uncertainties. In general, $R$ has the same trend i.e. the higher the water content of the seabed sediment, the smaller is $R$, but a distinction between subglacial unconsolidated or dilated till is not possible.

### 2.8  Determination of the source amplitude $A_0$

To determine $A_0$ we used the method described by Holland and Anandakrishnan (2009) as the direct path method, whereby the amplitudes of primary reflections of geophone pairs with a travel path ratio of 2 are compared. It was not possible to employ the alternative multiple bounce method (Smith, 1997), as the primary multiple is hardly visible in the data. Assuming $\alpha$ does





**Table 2.** Ranges of $R$ for different media contrasts at normal incidence

| Media contrasts with ice: | $R$ min. | $R$ max. |
|---|---|---|
| Lithified sediments/bedrock | 0.30 | 0.67 |
| Consolidated sediments | −0.05 | 0.18 |
| Unconsolidated sediments | −0.08 | 0.03 |
| Dilatant till | −0.11 | 0.00 |
| Seawater | −0.42 | −0.39 |
| Media contrasts with seawater: | $R$ min. | $R$ max. |
| Lithified sediments/bedrock | 0.62 | 0.67 |
| Consolidated sediments | 0.35 | 0.55 |
| Unconsolidated sediments | 0.28 | 0.41 |
| Dilatant till | 0.25 | 0.38 |

not change over the travel path, $A_0$ can be calculated. As our geophones are vertically orientated and the direct wave is a diving wave (a continuously refracted wave due to the continuous densification of the firn pack (Schlegel et al., 2019)), we used pairs of traces at larger offsets (97 m and larger) from the source which causes the ray-path of the diving wave to arrive at angles closer to normal incidence.

The detonating cord, placed in front of and parallel to the streamer, makes the source directional. This creates a wave front spreading cylindrically, perpendicular to the detonating cord orientation and semi-spherical at the ends of the cord. The cylindrically spreading wave front contains more energy and mostly agitates the subsurface whereas the spherically spreading wave front passes the streamer as a diving wave. This means we underestimate the source amplitude $A_0$ when using the direct path method.

At the ice–seawater interface, where the transition was abrupt, we determined $A_0$ by setting $R_{i-s} = -0.41$. At these transitions we know the acoustic impedance of the upper and lower media, namely ice and seawater, so here we can calibrate $R_{i-s}$. We refer to these shots as calibrated shots. Transitions from shelf ice to the seawater are not always acoustically abrupt, accreted ice or placelet ice may have formed at the base of the ice, giving (most often) a larger value for $R_{i-s}$. We considered 25 shots (21 at along-profile I and 4 at the across-profiles) of which eight (six at along-profile I and two at the across-profiles) had

an abrupt ice–seawater transition. From these eight shots we derived the source amplitude $A_0$ reliably by setting $R_{i-s} = -0.41$ and compared this with $A_0$ derived from the direct path method. The direct path method underestimates $A_0$ on average by a factor, the directivity factor, $D = 2.6$ ($2.1 \leq D \leq 3.1$). To compensate for the directivity of the source amplitude, we use $DA_0$ as the directionally compensated source amplitude as shown in Equation 2. The directionally compensated source amplitude has thus 19% accuracy.





Using $DA_0$ at five shots resulted in $R_{i-s} < -0.41$. As we assumed $R_{i-s} = -0.41$ is the smallest possible value for $R$, we set $R_{i-s} = -0.41$ at these shots and also refer to these as calibrated shots. With these additional calibrated shots we have a total of 13 calibrated shots (10 at along-profile I and 3 at the across-profiles) of the considered 25 shots.

Based on the noise level preceding the primary reflection of the bed or seabed, we determined $A_1$ with 7% accuracy. This
means we determined $R_{s-b}$ of the calibrated shots with 7% accuracy. Of the remaining 12 uncalibrated shots, where the directionally compensated source amplitude $DA_0$ has 19% accuracy, $R_{i-b}$ and $R_{s-b}$ could be determined with 32% accuracy.

## 3   Results

### 3.1   Seabed artefacts

In the following we will present time migrated and depth converted profiles. An exception will be the comparison of along-
profile II with along-profile I, where we will present time migrated profiles only. Time migrated sections are not suitable to unravel the subglacial structure of the seafloor when the ice shelf thickness shows significant variability over short distances. This is the case around the basal channel where the base of the ice shelf has significant topography and the seafloor is fairly flat. As the seawater is a low velocity layer ($V_p = 1425$ m/s) in comparison to ice ($V_p = 3750$ m/s), the thickness variation of the ice shelf causes significant time variation in the seafloor returns. The time migrated profiles show an apparent topography
(an almost mirrored version of the topography of the base of the ice shelf) in the seafloor caused by the different time delays of the ice shelf thickness. To derive the correct subsurface structure it is thus important to convert the migrated seismic profiles to depth. In general this works quite well but especially below the steep flanks of the basal channel the seawater–seafloor contact can not be properly recovered. In other words, the structure of the seafloor is influenced by the topography of the ice shelf.

### 3.2   Seismic along-profile I

The top of Figure 2 shows the 43.5 km long seismic along-profile I crossing the GL at shot point (SP) 26. The bottom of Figure 2 shows its schematic lay out with calculated values for $R$. There are ten locations where $R_{i-s} = -0.41$, six where the ice–seawater transition was abrupt and we calibrated $R_{i-s} = -0.41$ and four locations, SP 208, 209, 231 and 273, where $R_{i-s} < -0.41$ and was set $R_{i-s} = -0.41$. Based on the topography, structure and the reflectivity of the ice-base contact and seabed contact, we distinguish four intervals in along-profile I.

Interval 1 is from SP 1 to SP 44. We see a flat bed in direct contact with overlying ice. The bed starts at 1350 mbsl at SP 1, rising to 1300 mbsl at SP 26 after which the bed stays at 1300 mbsl to SP 44. The polarity of the ice–bed contact of the grounded ice is positive from SP 1 to SP 26, after which the polarity changes to negative for the rest of the profile. The location of the polarity change, SP 26, corresponds to the position of the GL derived by interferometry. From SP 4 to SP 22 the reflection coefficient $R_{i-b}$ increases from 0.16 to 0.43. From SP 26 to SP 44, $R_{i-s}$ is negative and decreases from -0.14 at
SP 30 to -0.41 at SP 33. The ice is uncoupled but the thickness of the seawater column is too small to be made out.



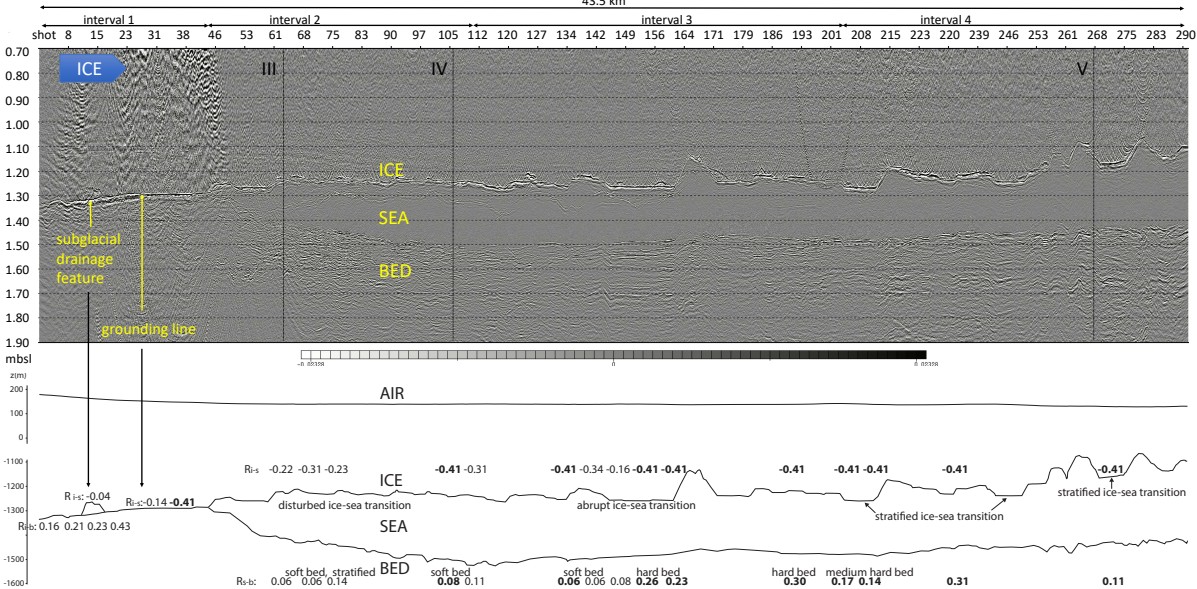

**Figure 2.** Top: Time migrated and depth converted (800 m to 2000 m) seismic along-profile I. The ice flow is from left to right, the $x$-axis indicates the shot point (SP) location and is divided in four intervals. The first 3.9 km are grounded ice then transitioning into an ice shelf. We identify the GL at SP 26 where the polarity of the ice base reflection switches from positive to negative. The crossings of the three across-profiles III, IV and V, are marked by the black dashed lines. Bottom: A schematic representation of the seismic profile above marking the boundaries of the ice surface and base and seabed. Clearly marked on the base of grounded ice is a subglacial drainage feature. The values of the calculated reflection coefficients are shown at their position. The bold numbers represent calibrated shots with $R_{i-s} = -0.41$ and $R_{s-b}$ has 7% accuracy, the normal numbers have 32% accuracy.

At the base of the grounded ice between SP 11 and SP 17 there is an elongated feature that appears to lie on a harder flat bed. It is approximately 50 m high and 1200 m long and has a small negative reflection coefficient ($R = -0.04$) at the ice bed contact. Our subsequent analysis shows this feature likely has evidence of subglacial drainage and hereafter will be referred to as the subglacial drainage feature.

5    Interval 2 is from SP 44 to SP 110. The ice–seawater contact, the base of the ice shelf, lies between 1220 to 1300 mbsl, has minor topography with anticlines 20 to 50 m above the surrounding base, and has $R_{i-s}$ values varying between $R_{i-s} = -0.22$ to $R_{i-s} = -0.41$. The ice–seawater contact is likely not abrupt, as the transition appears as an approximately 20 m sequence of discontinuous reflections. At SP 44 the ocean cavity deepens as the seafloor starts descending steeply to 1400 mbsl at SP 55 and then dips more gently to 1500 mbsl at SP 110. The polarity of the seafloor contact initially is small and positive but

10    occasionally, at SP 62, 66 and 73, negative. Between SP 55 and SP 110 the seafloor consists of a 200 m thick, transparent, stratified but disturbed sequence with $R_{s-b}$ varying between $0.06 \leq R_{s-b} \leq 0.14$. Downstream of SP 110 the character of the seafloor gradually changes to less stratified but still transparent.





Interval 3 lies between SP 110 and 204. The ice–seawater contact lies mostly between 1230 and 1260 mbsl and consists of (semi)horizontal terraces interchanged with anticlines 50 to 150 m above the surrounding base, reaching 1140 mbsl at SP 164. The ice–seawater contact is more abrupt, especially in the lower (semi) horizontal terraces (SP 145 to SP 160 and SP 171 to SP 180). The seafloor is transparent but less stratified with increasing acoustic hardness downstream from $R = 0.06$ at SP 130

to $R = 0.3$ at SP 191. This harder, less stratified but transparent bed can best be observed between SP 145 and SP 160.

Interval 4 lies downstream of SP 204. The seawater–ice contact now rises from 1270 to 1100 m, has less (semi)horizontal terraces and more anticlines. The ice–seawater transition at the terraces is no longer abrupt but a 20 to 30 m stratified sequence of parallel reflections. This is especially visible in the lower terraces (SP 204 to SP 212, SP 243 to SP 250 and SP 269 to SP 270). At this interval the ice–seawater transition has been set to $R = -0.41$ because the calculated $R < -0.41$. At the seafloor

$0.11 \leq R \leq 0.31$ is quite variable, larger then the second interval (SP 44–110) but not consistently high as in the third interval (SP 110–204).

### 3.3 Seismic along-profile II

We recorded along-profile II at the base of the surface channel. The profile is 10 km long starting 3.5 km downstream from the GL. Figure 3 compares the time migrated along-profile II with a time migrated part of along-profile I of the same length

(10 km). Both profiles intersect across-profiles III and IV at the same distance. At along-profile II the surface channel (on the surface) is not in perfect hydrostatic equilibrium, it is somewhat offset to the west with respect to the basal channel. Consequently, we recorded along-profile II over the western flank of the basal channel (Fig. 4). As a result the profile shows two events of the ice shelf–seawater contacts: one corresponds to the top of the basal channel, hereafter referred to as the roof, and the later one corresponds to the base west of the basal channel, hereafter referred to as the base of the ice shelf. For clarity

the roof and the base of the basal channel and seabed returns have been marked. Depth conversion of the time migrated stack would obscure the seabed.

Contrary to along-profile I (Fig. 3b) where the seabed consists of a 200 m thick stratification sequence between SP 60 and 87, at along-profile II (Fig. 3a), the seabed shows little sign of a stratification sequence in the basal channel. There is approximately 50 m of stratified material in the seabed between SP 50 and 42.

### 3.4 Seismic across-profiles III, IV and V

Figure 4 shows the three time migrated and depth converted across-profiles downstream of the GL. At across-profile III we used charges in 5 m deep boreholes whereas across-profiles IV and V were recorded with detonating cord. The borehole charges produce a ghost with $5 - 7$ ms delay, not present when using detonating cord at the surface. In other words, the source wavelet of the borehole charges is longer as with detonating cord charges. As a result, the ice–seawater and seawater–seabed contacts

of across-profile III are not as well resolved (they appear more stratified) as for across-profiles IV and V.

The basal channel on all three across-profiles is terrace-shaped, especially on the western side where the channel has a mid- and a high-level roof, as indicated in Figure 4b. On across-profiles III and IV, the lower level of the ice shelf base lies at app



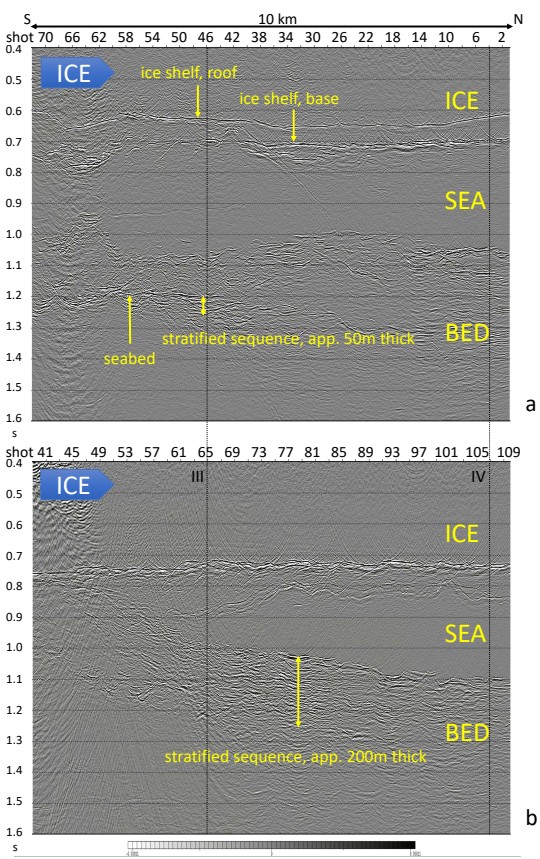

**Figure 3.** Time migrated along-profiles II (a) and part of I (b). Both profiles are 10 km long, starting 4 km upstream of across-profile III. Along-profile II is recorded on the western flank of the sub-ice channel (Fig. 4) and as a result has two events from ice–seawater reflections (see arrows). (a) The first one at 0.63 s (marked with arrow) represents the top of the sub-ice channel (channel roof) and the second one at 0.7 s represents the ice shelf west of the sub-ice channel (base). The sub-shelf structure is influenced by the seawater column thickness but we can pinpoint the seabed in the basal channel (yellow arrow) based on across-profiles III and IV, marked by the black dashed lines. The seabed has some stratification but this sequence is significantly less thick then along-profile I shows. (b) A 10 km time migrated section of along-profile I. In order to compare the section with (a) it has not been depth converted as in Figure 2.

1330 mbsl and the roof of the basal channel, at 1050 mbsl. At across-profile V the character of the channel roof is more rounded but the terraces can still be made out. The base lies at 1250 mbsl and the high-level roof of the sub-shelf channel at 920 mbsl.

The seabed at the across-profiles lies between 1450 mbsl and 1500 mbsl, where we have to keep in mind that the migration and depth conversion (and thus topography) of the seabed under the steeper flanks of the sub-shelf channel, is not correct.

5   Especially across-profiles IV and V have a flat seabed that only shows some "apparent" topography under the steeper flanks of the sub-shelf channel. The across-profiles show a thicker stratified sequence under the sub-ice channel which extends to the eastern side on across-profile III.

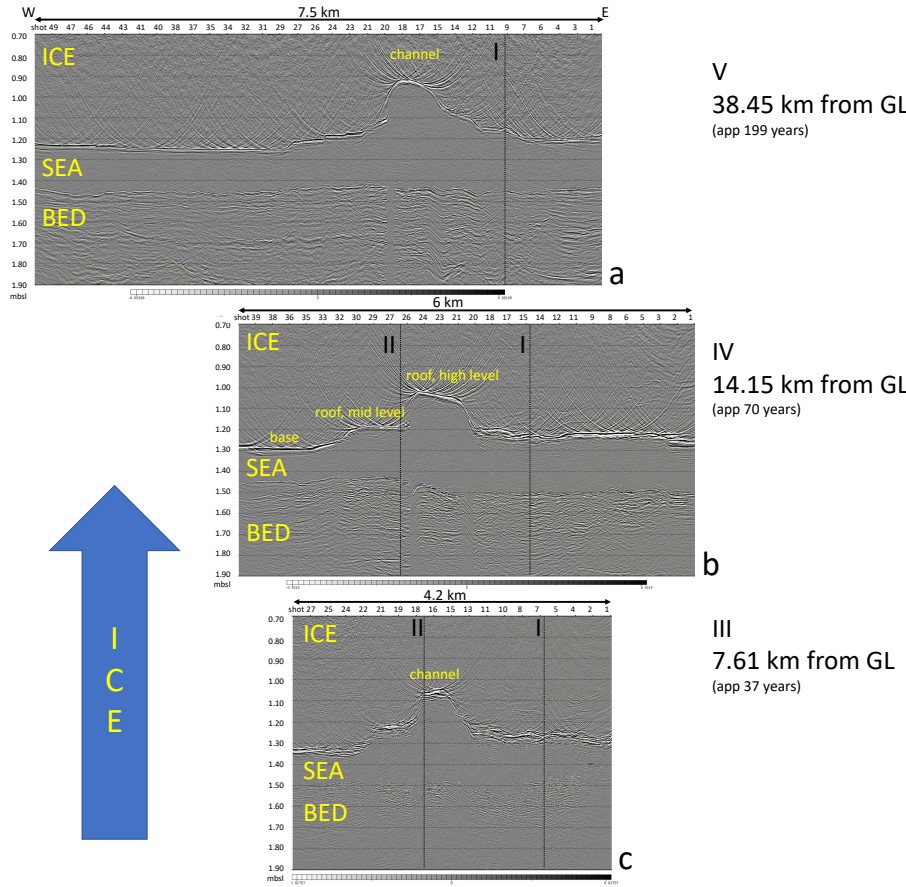

**Figure 4.** Three across-profiles in along-flow sequence to show the development of the sub-ice channel with the most upstream profile, along-profile III, at the bottom. The ice flow sequence is from bottom to top. The crossings of the along-profiles I and II are marked by the black dashed lines. The distance from the GL and the time lap for the ice to reach this distance are mentioned on the right side. The across-profiles have been aligned across-flow with respect to the most westerly flow line of Figure 1a. As a result the crossings of the along-profiles (black dashed lines) shift slightly westward with flow. (a) The most downstream across-profile V, with the high-level roof of the channel marked. (b) Across-profile IV, for clarity the high and mid-level roof of the channel as well as the base are marked. (c) The most upstream across-profile III.

To see the development of the lateral position and the changing geometry of the basal channel, Figure 5 compares three 3.75 km long schematic sections derived from the seismic across-profiles in sets of two (section III with IV and section IV with V). From across-profile III to IV (Fig. 5b), the terrace shaped multi leveled roof of the channel form stays preserved but widens from some 780 m to 920 m, rather than that the ice shelf thickness changes. The flanks of the basal channel become steeper but the height does not change noticeably although there may be some lowering in the center of the basal channel. From across-profile IV to V (Fig. 5a), the terrace shape of the basal channel becomes less pronounced and the base is shallower. The





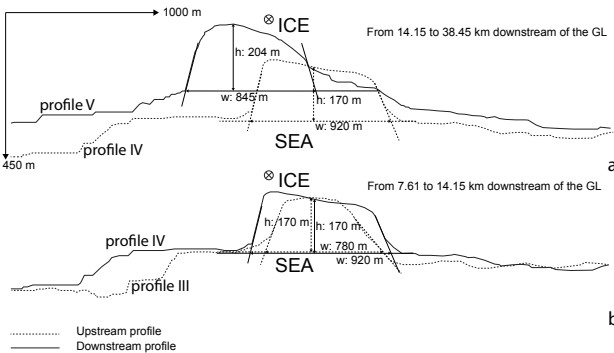

**Figure 5.** A schematic comparison of the ice shelf development around the basal channel derived from the three seismic across-profiles in Figure 4. The ice flow is into the page. All 3.75 km long sections have been lined up against the most western flow line of Fig.1a. Vertically, they are positioned with respect to the ice shelf surface so that the ice thickness can be compared. (a) Comparing across-profile IV (dashed line) with V (continuous line). (b) Comparing across-profile III (dashed line) with IV (continuous line).

flat roof of the channel becomes more rounded and the lower level roof on the western side, less pronounced. As the base is shallower, the ice shelf at across-profile V is thinner. Based on the flow line orientation the channel has moved westward in the downstream direction.

### 3.5 Basal channel characteristics in ten radar across-profiles

We selected ten across-profiles separated by 2.6 km in along-flow direction (Fig. 6). These track the basal channel and the landform shaping the channel upstream of the GL. The ~3.75 km long radar profiles are shown in light blue and the basal channel and the landform shaping it upstream of the GL (hereafter referred to as landform) in dark blue in Figure 1d. The radar profiles are rotated 5° SW–NE with respect to the seismic across-profiles. Profile 10 lies 10.85 km downstream of the GL, profile 6 is at the GL, and profile 1 lies 12.83 km upstream of the GL. The basal reflection of the ice is marked in blue.

The resolution of the radar data is not as good as of the seismic data, so the shape of the basal channel can not be reconstructed as well. Nevertheless, we can track the basal channel and, upstream of the GL, landform up to radar profile 3, at least 7.7 km upstream of the GL after which the landform becomes indistinguishable from the bed.

## 4   Discussion

### 4.1   The grounding line position

In Figure 1c three GLs are shown: in red MOA (Scambos et al., 2007), in blue ASAID (Bindschadler et al., 2011) and in yellow a GL we derived from interferometry. At along-profile I the polarity of the basal reflection switches from positive to permanently negative in downstream direction at SP 26. As negative polarity indicates the presence of water at the base, this confirms the ice uncouples from the bed. Between SP 30 and 33 we see a decrease in $R$ from $R_{i-s} = -0.14$ to $R_{i-s} = -0.41$

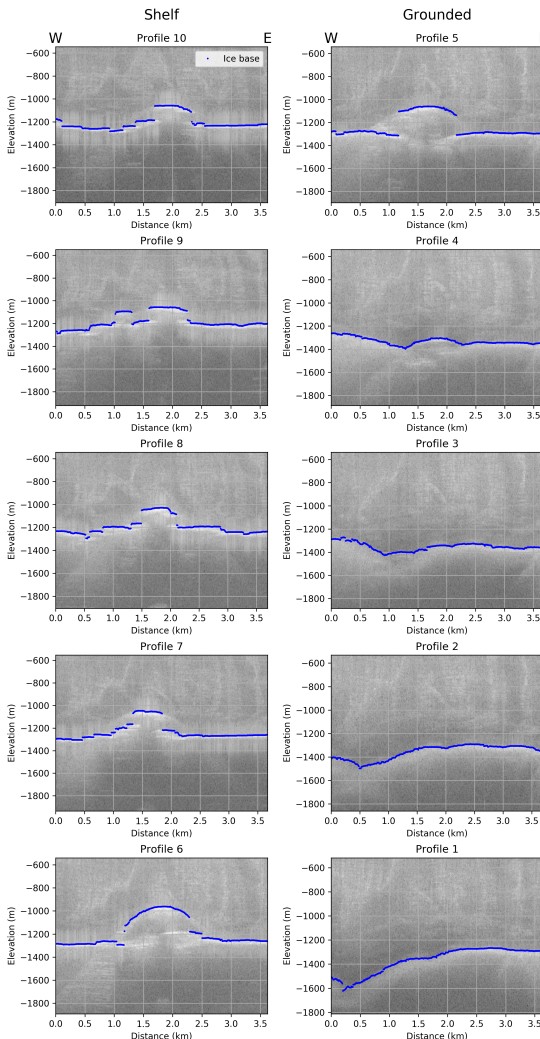

**Figure 6.** Return power, grey-scaled and depth converted radar profiles 1 to 10. Figure 1d shows their position. The ice flow direction is from bottom to top, starting with profile 1 (upstream) at the lower right corner up to 10 (downstream) in the upper left corner. The basal reflection (seawater and bed) of the ice is marked in blue. At profiles 1 to 5 the ice is grounded, profile 6 is at the grounding line and at profiles 7 to 10 the ice is floating.





which is probably caused by an increase in water content in the subglacial bed. At SP 33 the ice is in contact with the seawater. We recorded SP 26 at 17:06 UTC, January 1, 2017. According to five GPS stations 13 km downstream from the GL, this is 3.5 hrs after high tide at which time there was 1.5 m additional uplift on a tidal range of 2.6 m. As SP 26 corresponds well with the GL derived from interferometry and not with the GL derived from the surface slope (MOA) we refer to the GL as the one

provided by interferometry in the remainder of the paper.

## 4.2 The structure of the ice shelf and ocean cavity

Looking at the structure of the ice sheet at the GL area of along-profile I (Fig. 2), we see an almost constant ice thickness gradient when the ice, initially flowing over a flat bed, passes the GL. Unlike the classic picture of the sheet–shelf transition, where rapid ice shelf thinning close to the GL causes a steeply rising ice shelf base, the sheet–shelf transition of SFG seems to

be a mirrored version of this: SFG has a steeply descending seafloor at SP 44 and an almost constant ice thickness downstream of the GL. This steeply descending seafloor, the onset of the ocean cavity, probably determines the GL position. The lacking of an ocean cavity at the flat ice shelf base upstream of SP 44 confirms this.

Generally we would expect the highest melt rates at the deepest part of the ice shelf, the GL, as that is where the melting point is lowest due to the pressure effect of the ocean. The constrained ice flow, confirmed by the parallel flow lines (Fig. 1a)

and the flat ice shelf base, allow us to use the ice thickness gradient as a first order approximation for basal melt (Fig. 2). As the base of the ice shelf is initially flat, any topography in the base is likely to be caused by basal melt. The constant ice thickness gradient of the ice passing the GL suggests there is little basal melting at the GL of SFG.

Once in contact with the ocean cavity at SP 44, there is some basal melting as the base of the ice shelf has some topography but this increases in downstream direction as we see an increase in the number of and the magnitude of anticlines in the ice

base. Interval 2 has some topography but small compared to interval 3 and 4. Interval 3 has one pronounced anticline at SP 164 interchanged with lower terraces and interval 4 has several pronounced anticlines, at SP 215 and downstream of SP 250, interchanged with lower terraces. Dutrieux et al. (2014) observed a similar ice shelf geometry and attributed the terrace shaped structure to a steplike thermohaline ocean structure causing organized melting.

That there is little melting at the GL increasing in downstream direction is also confirmed by the seismic across-profiles of

Figure 5b. The ice shelf base of across-profiles III and IV does not change much in depth but the basal channel itself widens. At Pine Island Glacier this channel widening was ascribed to ice dynamics, i.e. convergence at ice shelf surface and divergence at ice shelf base (Dutrieux et al., 2013; Vaughan et al., 2012). At SFG we observed no noticeable ice convergence at the surface channel, at least not distinguishable from the noise level. Between across-profiles IV and V we observe a general thinning of the ice shelf, both, above the sub-ice channel and outside of it. Across-profile V crosses along-profile I in interval 4 where there

is increased basal melting of the ice shelf.





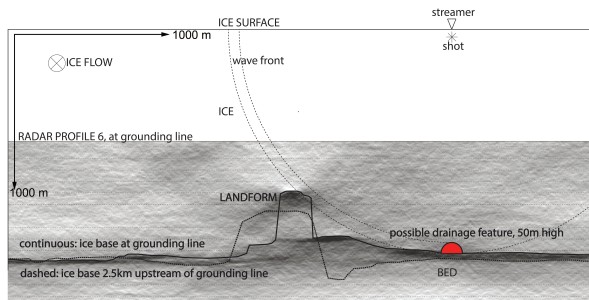

**Figure 7.** Cross section of the seismic recording geometry of along-flow profile I during shots 11 to 17. The shooting direction and ice-flow direction are into the page, perpendicular to the cross section. The continuous and dashed black lines form the ice base derived from radar profiles 5 (continuous and greyscale image in figure) at the GL, and 6 (dashed) 2.6 km upstream from the GL. The landform at off-nadir direction is pointed out. The red semicircle represents the dimensions of a hypothetical subglacial drainage feature at nadir (50 m high, 1200 m long). The circular wave front reaches the base at nadir and the landform shaping the shelf channel at the same time.

### 4.3 Characteristics of the subglacial drainage feature

At along-profile I, below the grounded ice between SP 11 and 17, the subglacial drainage feature appears to lie on a harder flat bed. It is unclear whether the subglacial drainage feature lies at nadir or off-nadir direction. Depending on this we have two possible interpretations:

1) If the seismic event is from off-nadir, we most likely recorded the top of the landform shown in Fig. 7. The schematic cross section shows the recording geometry of SP 11 to 17 from along-flow profile I. The ice base, showing the landform, comes from dashed profile 5, 2.6 km upstream of the GL, and continuous to radar profile 6 at the grounding line. The spherical wave front, represented by the dashed circular line, reaches the off-nadir landform before the bed at nadir. In this scenario reflection coefficient $R_{i-b} = -0.04$ but $\theta = 45°$, so we need to consider the angle dependency of $R$. Although the amplitude

may be 32% accurate, the polarity is not affected by this uncertainty. For subglacial consolidated to unconsolidated sediments, $R < 0$ becomes smaller with increasing $\theta$. So, if the event is from off-nadir, then the the landform consists of sediments having some degree of consolidation.

2) If the seismic event is from nadir, it would be a separate drainage feature on a hard bed, 1200 m long and approximately 50 m high, its dimensions represented by the red semi-circle in Fig. 7. The reflection coefficient $R_{i-b} = -0.04$ at normal

incidence indicates that the subglacial material would consist of unconsolidated water containing sediments. So if originating from nadir, the subglacial drainage feature most likely is a stand alone subglacial conduit on a hard flat bed transporting wet sediments which are deposited at the downsloping seafloor, and the seismic event is not related to the landform and surface channel.

Both interpretations have consequences for the seismic acoustic velocity model and the resulting profile structure, because

the subglacial drainage feature has a significantly lower $V_P$ (1900 m/s) than ice (3750 m/s). If interpretation 1 is correct, i.e.

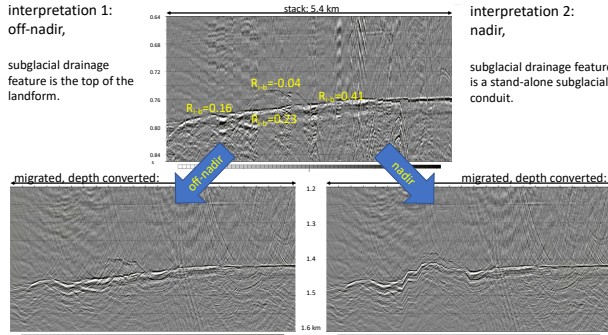

**Figure 8.** Top: A zoom of the the subglacial drainage feature on a hard flat bed of the unmigrated along-profile I, also named stack. The negative polarity along the subglacial drainage feature clearly shows up. Bottom: Two time migrated and depth converted zooms, showing the effect on the underlying hard bed of the subglacial drainage feature, depending on wether we use a two-layer velocity model when the feature is off-nadir direction (lower left: interpretation 1), or a three-layer velocity model if the feature is nadir (lower right: interpretation 2).

the subglacial drainage feature is off-nadir and is in fact the top of the landform, ice alone covers a hard flat bed. In that case the velocity model consists two layers: a top layer representing ice and a lower layer representing the subglacial flat hard bed. If interpretation 2 is correct, i.e. the subglacial drainage feature is nadir and a stand alone conduit on a hard flat bed, there would be a low velocity layer (the subglacial drainage feature) in between the ice and the hard flat bed. In that case the velocity
model has three layers: a top layer representing ice, a second layer representing unconsolidated, water containing sediments and a third layer representing the subglacial flat hard bed.

The two versions of the time migrated, depth converted profiles using either velocity model are shown in Figure 8. If the subglacial drainage feature is off-nadir and in fact the top of the landform, we use the two-layer velocity model. The bed of the time migrated, depth converted profile, stays flat which agrees with both radar across-profiles that show no topography below
along-profile I. If the subglacial drainage feature lies nadir, and we are dealing with a stand alone subglacial conduit, we use the belonging three-layer velocity model. The hard bed of time migrated, depth converted profile is no longer flat. Below the subglacial conduit the hard bed has a local rise. On an otherwise featureless bed, this is possible, but coincidental. Next to this, both radar profiles are featureless apart from the off-nadir landform. A 50 m high, 1200 m long subglacial conduit, as represented by the red semi-circle, would show up on the radar profiles unless this feature is so local that it does not cross the
2.6 km separated radar profiles. That is possible but again, would be coincidental.

Because of these two argumentations we prefer interpretation 1: the subglacial drainage feature is off-nadir and is in fact the top of the landform. With $R_{i-b} = -0.04$ and $\theta = 45°$ the bed most likely consists of sediments but the degree of consolidation is difficult to determine. This conclusion is based on the reflectivity of the off-nadir reflection of SP 11–17. As we lack a proper seismic across-profile over the landform, we can not retrieve its full structure. It may be that only the surface of the eastern side
of the landform causing the reflection, consists of sediments and the deeper part bedrock.





## 4.4 Properties of the seabed

At along-profile I, we determined $R$ at 21 locations over 43.5 km starting when the ice is grounded. Due to the directionality of the detonating cord used as seismic source, we determined $R$ within 32% accuracy at 11 of the 21 shots, where $R$ could not be calibrated against the ice–seawater transition of the ice shelf. At these places, we use the trend or polarity of $R$ for

interpretation rather than the actual value.

At all the places where the ice–seawater contrast is not abrupt, the propagating p-wave, traveling from the ice shelf into the seawater, encounters a series of transmissions and reflections rather than a single transmission. The total amplitude loss over such a stratified ice–seawater contrast is probably larger then our assumed transition of Equation 6. At these place we most likely underestimate $R_{s-b}$.

While still grounded, we see an increase in $R_{i-b}$ from 0.16 to 0.41 over a 3.9 km distance. As we find it less likely that the subglacial material changes drastically over this short along-flow interval and the acoustic impedance $Z$ of the ice is constant, we attribute this steady increase in acoustic impedance of the subglacial material potentially to increasing compaction, as has been observed by Christianson et al. (2013). We interpret the bed of the grounded ice thus as subglacial till.

Once the ice passed the GL at SP 26, the uncoupled ice stays in contact with its flat base down to SP 44 when the seabed

starts to descend steeply. The seabed of the ocean cavity (downstream, of SP 44) consists of two distinguishable environments. At interval 2, close to the GL, we have a 200 m thick stratified and disturbed sequence that changes into an undisturbed less stratified seabed at interval 3 and 4. The disturbed stratified sequence close to the GL, having smaller values for $R_{s-b}$, consists of softer, porous material then the undisturbed, less stratified seabed. We interpret this disturbed stratified sequence of the second interval as grounding line deposits consisting of subglacial terrestrial sediments. The softness of the seabed is confirmed

by the occasionally negative polarity at the steeper downslope of the seabed. The undisturbed, less stratified seabed, generally has higher values for $R_{s-b}$, indicating a harder material. This is particularly clear in interval 3, where $0.23 \leq R_{s-b} \leq 0.30$. The seabed structure representative for this second type of environment is most clearly visible between SP 146 and 161, where the flat featureless ice–sea contact does not influence the seabed topography. In interval 4 we calculated lower values for $R_{s-b}$ but these low values all have a stratified ice–seawater contact above them and so the amplitude loss at the ice–sea transition may be

larger then is accounted for. The stratified contact probably causes a larger energy loss than the assumed abrupt ice–seawater transition of Equation 6 accounts for and probably underestimates $R_{s-b}$. Based on $R_{s-b} \approx 0.31$, we believe we are dealing with consolidated sediments but can not exclude bedrock. In any case, this part of the seafloor has properties of an eroded surface, where softer deposits are missing. It could have been created during periods of higher ice-dynamic activity, e.g. during one or several advances of SFG during the last glacial into LGM positions of maximum advance.

## 4.5 The subglacial hydrological interpretation

The landform and basal channel lie on the western side at the western shear margin of SFG. From the radar profiles we can track the landform at least 7.7 km upstream of the GL up to radar profile 3. The landform consists of sediments having some degree of consolidation on its eastern side, but may have a hard rock core as suggested by Jeofry et al. (2018). The ocean





cavity close to the GL and downstream of the landform, has a 200 m thick soft stratified, disturbed sedimentation sequence. Sub-shelf noble gas samples (Huhn et al., 2018) suggest subglacial water influx at SFG and modeling of subglacial water routing places the water influx at the western and eastern sides of SFG. Taking the evidence together we conclude the landform is hosting the transport of sediments that are deposited in the ocean cavity close to the GL. Coming back to the classification

of surface channels by Alley et al. (2016), we are dealing with a type (2) surface channel; a subglacially sourced channel (the landform) that intersects with the GL and coincides with modeled subglacial water drainage. Basal channels do often form in shear margins (Alley et al., 2019) and the location of this basal channel coincides with the western shear margin of SFG but it is the landform that initiates this basal channel. Possibly, shear margins facilitate the formation of landforms hosting the subglacial transport of sediments by water routing.

What may seem surprising is that we see a much thicker stratified sediment sequence at along-profile I than at along-profile II, which suggests the subglacial drainage is closer to along-profile I. This is not necessarily the case, as along-profile II starts 3.5 km downstream of the GL so we do not know the seabed structure at the GL. What we do see is more stratification in all across-profiles below the sub-shelf channel than outside of it and that this stratification extends to the eastern side of across-profile III. Keeping in mind that the off-nadir reflection of the subglacial drainage feature comes from its eastern wall

of the landform, we conclude the landform hosts subglacial drainage on its eastern side that caused the disturbed stratified sedimentation sequence at the GL of along-profile I.

At radar profiles 1 and 2 the landform can no longer be identified. However, if we follow the flow line upstream connecting the basal channel and landform (the red line in Fig. 1a and d), it connects to an elongated ice surface feature visible in the REMA data (Howat et al., 2019). We do not have radar data to confirm the shape of the bed but we hypothesize that the

landform continues further upstream here. If this is indeed the case, the question arises why the landform can not be tracked at radar profiles 9 and 10. Our speculation is that the landform is present at radar profiles 9 and 10 but buried in subglacial material, transported by a subglacial channelized system. As such the landform cannot be detected by the airborne radar profiles 9 and 10.

## 5 Conclusions

We investigated the characteristics of a basal channel across the grounding line of the Support Force Glacier. Our observations do confirm with the categorization of Alley et al. (2019), i.e. the basal channel conincides with the western shear margin. However, rather than the channel being formed by the surface depression in the shear zone of the grounded ice and adjusting to the hydrostatic equilibrium once the ice is afloat, we find that the channel originates at a subglacial landform, which agrees Jeofry et al. (2018). On its eastern side the landform consists of sediments, but we cannot conclude whether a deeper hard rock

core is also present.

The landform hosts a channelized subglacial drainage, which transports sediments downstream. These are deposited just seaward of the grounding line, where we identified a 200 m thick soft, disturbed, stratified sedimentation sequence at the seafloor. Further downstream the seafloor consists of harder, undisturbed and less stratified consolidated sediments, possibly



also bedrock. We attribute these two units to originate from different development phases: whereas the harder sequence is potentially a left-over from a farther advanced grounding line, e.g. coming along with stronger erosion during advances into the LGM, the softer sediment sequence seems to be the result of comparatively recent post-LGM and Holocene grounding line depositions.

Apart from the channel and individual anticlines, the base of the ice shelf downstream of the grounding line is relatively flat and almost horizontal, indicating that basal melt rates are relatively low. We attribute the observed widening of the basal channel to melting along its flanks, which we also observe at the flanks of anticlines in the ice shelf base. The melting increases further in downstream direction. To date it is unlikely that warmer water is already in the cavity near the grounding line. But even if it was, the geometry of the steeply descending seafloor downstream of the grounding line would limit the direct contact

of potentially warmer water with the ice shelf base, thus limiting basal melt rates, unless the cavity was fully flooded with warmer water. This is in contrast to the typically envisaged geometry and ice–ocean interaction at grounding lines, which often envisage a steeply rising ice shelf base just downstream of grounding lines, where circulation is dominated by the ice pump mechanism. With our improved characterisation of the grounding line area of SFG, future modelling studies should investigate how this area might react differently to the presence of warm deep water in the cavity than for instance the glaciers in the

Amundsen Sea Embayment region and thus quantify the role of the seafloor geometry.

*Acknowledgements.* The Filchner Ice Shelf Project (FISP) was funded by the AWI Strategy Fund. The Grounding Line Location (GLL) product was provided by DLR via ESA CCI Antarctic Ice Sheet. Hugh Corr was supported by the UK Natural Environment Research Council large grant "Ice shelves in a warming world: Filchner Ice Shelf System" (NE/L013770/1). We thank the field guides Dave Routledge and Bradley Morrell for their unwavering guidance and support. We also thank BAS for their logistic support and hospitality. A special thanks

goes to the pilots delivering the hardware at the right time at the right remote place. Without them, this survey would have been impossible.



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
