# Peer review of "Evidence for a grounding line fan at the onset of a basal channel under the ice shelf of Support Force Glacier, Antarctica, revealed by reflection seismics."

_The Cryosphere, 2020_

## Referee Comment (RC1) · Adam Booth (Referee) · 23 Jun 2020

Dear authors,

I really enjoyed reading your manuscript - I'm very sorry that it has taken me so long to review it.

General comments:

The paper explores a little-studied feature in the base of Antarctic ice shelves, but one

that is nonetheless important for ice shelf stability and sub ice-shelf circulation. The study also indicates subglacial sediment transport. The data acquisition and processing is of a high standard and the imaging is unambiguous. Most of the comments in the attachment are suggestions for grammatical corrections or improvements for readability; some of the paragraphs seem a little rushed, and some sentences aren't as easily understood as I think they might be. I also have some more substantive requests for clarification of a few points, as mentioned below, but I don't think that these are major issues. Overall, I think that the paper is a nice contribution to the understanding of ice shelves and their dynamics.

Best regards, Adam Booth

Specific comments:

Abstract - I found this a little qualitative, and maybe some hints at the dimensions of the subglacial channel could be useful?

P2 L7-8 - similar to my comment in the abstract, maybe some quantitative detail about the geometry of surface channels could be useful?

P3 L12 - 'active source' seismic

P3 L16 - if its important to mention the date of the radar acquisitions, do so for the seismic too; I'm not sure you say when the data were acquired at any point in the manuscript.

Figure 1. Add a distance scale for the arrow lengths in (a). I also wonder if the figure would benefit from some textual geographic labels (e.g., the landmarks and features mentioned in the intro?).

Section 2.5 - again, I'd mention the acquisition date here.

P6 L3 - define p as compressional (equally, S as shear when it comes up later). Additionally, you occasionally swap between "P wave" and "p wave". Be consistent.

P6 L4 - your sample rate is actually a sample interval.

P6 L5 - consider putting the manufacturer of the GEODE system... something like "four Geometrics GEODE units"?

P6 L7 - should II be III, given the information in Table 1?

P6 L10 - I think your description of the acquisition (particularly for the long offset gathers, but maybe also for the profiles) might benefit from a schematic diagram of the survey.

P7 L9 - define R(theta) at the start of this paragraph, otherwise Reflectivity and the symbol in Equation (2) is undefined.

P7 L11 - it's unclear how Vp, Vs and rho relate to the primary reflectivity, in the way that you have described it here. I would consider splitting this sentence, explaining how reflectivity is defined by contrasts in these quantities, and then introducing Equation (2).

P7 L14 - just give the Section number explicitly, rather than the cumbersome "determination of A0 subsection".

P7 L20 - give references for the parameter ranges you use in Table 2, to provide you with reflectivity ranges.

Table 2 - explicitly state which material would have subscript 1 and 2 (i.e., which is above and below the interface). You might also consider defining the impedances as well as the Reflection coefficients?

P9 L15 - the abrupt transition is, of course, only abrupt on the wavelength scale of your wavelet; maybe simply adding "at the XX m scale of vertical resolution" in here?

P9 L19 - here, and throughout, I think you're mis-using the term 'accuracy'. If something has 19% accuracy then it is very poor indeed! Do you mean 19% uncertainty, or "accuracy better than 81%", or suchlike? If I'm correct, that you're mis-using this term, make sure that other instances of accuracy are checked too. In this specific case, it's

also not clear to me how the numbers above end up giving this accuracy.

Section 3.1, header - I'm not sure that these are 'artefacts' as such, which I'd consider more to be residual effects of processing (e.g., migration smiles). I'd suggest that this subsection is retitled "Seabed depth conversion"?

Figure 2 - this is a nice figure, but I'd suggest that the other figures in the section are given the same interpretation panels - it's unclear why you'd only provide it for this one. It might also be good to show an enlarged section of the subglacial feature; I know it features in other figures, but I really couldn't see it here.

Figure 2 caption - "switches from positive to negative" - how do you define what is positive and negative polarity? You might just be better saying "changes polarity".

P11 L3 - it seems a little premature to be referring to this feature as a 'drainage' feature. It's only in the Discussion where you start to present the evidence for this, based on previous work. At the moment, it is a subglacial feature, but it's impossible to know it's a 'drainage' feature from the seismic results alone. I'd suggest that you remove 'drainage' at this point in the manuscript.

P11 L7-8 - is the gradational, rather than abrupt, transition the reason why you see the deviation to smaller-magnitude reflectivity? Is it worth making this comment explicitly?

P12 L17 - The sentence "Consequently..." makes it sound like you did this deliberately, whereas I don't think you did at all! I'd rephrase this as "Consequently, along-profile II samples the west flank of the channel rather than its crest, and therefore complicates the recorded seismic response."

P12 L17-19 - It took me quite some time to recognise what was going on with the appearance of the 'double bed' in Profile II and Figure 3. I think you need to explain the geometry more clearly, and explain that you have these two laterally-offset reflectors within a Fresnel zone of each other. I also think it would be helped if you presented Profiles III, IV and V first - they don't have to come in numerical order. That said, I do

wonder if Profile II adds much to the interpretation - you don't really refer to it later in the manuscript, and it's clearly not acquired in the most ideal location (not that I blame you, of course, it happens!!). Maybe it should be relegated to supporting material?

P12 L20-21 - Why would depth conversion obscure the sea bed?

Figure 4 - these seismic images are lovely :)

P15 L17-18 - with the likely complex pattern of reflectivity close to the uncoupling, I'm not sure you can say that the polarity 'confirms' the presence of water - but it might certainly support or imply it.

Figure 6 - it might be good to include a refresher of the location map?

Section 4.3 - again, I'm not sure that 'drainage' is yet appropriate in this section header.

P18 L11-12 - Given that the landform likely represents a diffracting point rather than a specular reflection, I'm not sure that reflectivity calculations hold. I agree with your geometric arguments and think that you do a good set of analyses here, and I think that the reflectivity argument is in any case superfluous. I'm not sure what the reflection coefficient equation would be for this; I think you can likely speculate that the amplitude appears weaker than surrounding reflections, but the quantitative assessment might be an over-interpretation. (this comes back on P19 L17-18).

P20 L4 - define 'trend'. Do you mean the magnitude? As in, you're interpreting based on indicative reflectivities rather than a fully-quantitative asessment?

P20 L16 - I wonder if the terms 'disturbed' and 'undisturbed' imply a process rather than a geometry? As in, the implication that the sediment has been disturbed by something (e.g., ocean currents). Of course, this might be the case, but as an indicator of simple geometries then I think that 'stratified' and 'unstratified' of 'homogeneous' might be less weighted?

P22 L5-6 - I think flat and horizontal might be the same thing? The difference here
might be in the terms 'planar' and horizontal.

P22 L19 - I agree with you, Bradley Morrell is great!

Technical issues:

There are many grammatical issues which I have flagged up in the attached manuscript. These are flagged up, and suggestions made for alternative wording; all of the comments above are also included.

Please also note the supplement to this comment:
https://tc.copernicus.org/preprints/tc-2020-54/tc-2020-54-RC1-supplement.pdf

**Supplement:**

[revised manuscript text omitted]

---

## Referee Comment (RC2) · Neil Ross (Referee) · 3 Jul 2020

This is a well reported high-resolution seismic reflection survey targeting the form and physical properties of an ice shelf channel and the sub-seafloor sediments beneath it at Support Force Glacier (SFG), East Antarctica. The seismic analysis is supported by some airborne ice-penetrating radar data. The methods are sound and detailed, the description and presentation of the data is reasonably good, and the science is high-quality and of potential interest to the readership of TC. The data are hard won field geophysical data from a remote part of the Filchner Ice Shelf, Antarctica and

certainly deserve to be published in some form. I do, however, have some serious concerns about the way the data are 'pitched' and argued in the current version of the manuscript. Specifically, I am very unconvinced by the association between the ice shelf channel, the "subglacial landform", and the sub-seafloor sedimentary structures, and the argument for sediment transport subsequently developed.

General comments 1. The manuscript does not engage at all with glacial-geological literature relevant to glaciomarine processes and sediment deposition in a grounding zone and ice shelf environments. Such process literature is key to understanding the sedimentary structures imaged in the seismic data. Without reference to such literature you cannot make the link between the present-day ice shelf channel and the sediments beneath the sea floor. Though there are clearly more modern literature available, a good place to start would be David Drewry's textbook on Glacial Geological Processes (1986). What are the processes that the seismic observations of the cavity and the subsea sediments give insight into? What might be the glaciological processes that determine sedimentation in ice shelf cavities and at grounding lines?

2. I am not, at present, convinced that the observations of the stratified sediment beneath the ice shelf channel have any bearing on the ice shelf channel and modern-day "sediment transport" itself. The manuscript makes no convincing case that the sediments were deposited by present-day processes. These sediments could be much older than the ice shelf channel and may have absolutely no relationship with modern-day processes at the grounding line or beneath the ice shelf. The authors need to either (1) provide a much stronger justification for the direct link between the sediment and the modern-day glaciology (e.g. by using the literature on glaciomarine sedimentary processes I refer to above and/or better describing and presenting the data they report). It is not enough to simply say on page 21 that "we conclude the landform is hosting the transport of sediments that are deposited in the ocean cavity close to the GL" – what is the evidence?; or (2) reframe the paper so that it is a detailed characterisation of the form and physical properties of (a) the ice shelf channel; (b) the sub-shelf bathymetry;

and (c) the sub-bottom sediments of SFG, but doesn't link them directly. For what it is worth I think a manuscript describing '2' would be useful and worthwhile. We know so little about Support Force Glacier at present.

3. The assertion in the abstract and section 4.5 that the "landform is hosting the transport of sediments that are deposited in the ocean cavity close to the GL" (page 21) is very poorly supported by any evidence apart from the spatial coincidence between the landform (described in other parts of the manuscript, e.g. section 4.3, as a "subglacial drainage feature") and the ice shelf channel. What is the process that is being inferred here? In the conclusions it is suggested that the sediment transport is by subglacial meltwater, but that is not developed from a detailed, carefully constructed and coherent argument in the manuscript.

Please don't get me wrong, I think the manuscript is full of great data and interesting observations, but at present it seems to lack a clear focus, and many of the assertations are not fully thought through or developed from a process-oriented perspective. It's all very well to justify the landform identified in the ice-penetrating radar as being comprised of sediment from the seismic reflection coefficient analysis (though it is clear that the seismic data are not perfectly acquired to do this), but the manuscript makes some very large leaps from 'this is a subglacial sedimentary landform (with possibly a bedrock core)' to the stratified sediment offshore is the direct result of focused melting from the ice shelf channel. What is the process by which this happens? It needs to be justified.

An idealised conceptual model of the ice shelf cavity/grounding line/ice shelf channel processes and environments might be useful. See examples in Le Brocq 2013; Drews et al. 2017; and Jeofry et al. 2018 (Suppl Info). A conceptual model like these would really help pull together how the SFG system works and make the manuscript far more accessible to prospective readers. I suspect that developing one might also help the authors think through the processes and allow them to piece together a much more coherent argument and explanation for their observations.

[Figure]

Please note that there are a few typos and minor grammatical errors throughout the manuscript that will need correcting before publication. I have highlighted some below, but I have not been comprehensive in this.

Specific comments

Title: Support Force Glacier is East Antarctica, not West.

Title: Where is the evidence for "sediment transport" in the paper?

Addresses: 'Natural Environment Research Council' (i.e. not National Environmental).

Abstract (L1): "surface channels" not "flow stripes". Flow stripes are something different, and are not associated with basal ice shelf channels.

L5: "beneath" rather than "on"?

L6: "part" of what?

L8: "initiates" rather than "forms"?

L10: What is the justification for the 200 m thick sequence of sediments being "grounding line deposits" – what is the evidence and argument for this? At present the manuscript doesn't provide this.

L10: "the landform hosts the subglacial transport of sediments" – why not just "the landform is composed of sediments". That seems to be as far as you can take the interpretation of the seismic data as far as I can make out from the manuscript. There is no evidence for subglacial transport of sediments from the data presented.

L15: "shear margins" are introduced here, but figures 1a&1b suggest that the subglacial hydrological pathway does not correspond to the shear margin.

L4: surface expressions of basal ice shelf channels are not the same as flow stripes, and sometimes they 'jump' across flow stripes. They are surface features associated with linear depressions in the ice shelf surface.

L25-27: some misrepresentation of Jeofry et al. 2018 here – the hard rock landforms at Foundation Ice Stream (FIS) actually determine subglacial hydrological pathways & it is the basal water that forms the ice shelf channels, not the bedrock bumps per se. The marine landforms presented in Jeofry et al. 2018 did not "confirm" anything. That particular figure was simply included to demonstrate how common such hard bed landforms were offshore and presenting them as a plausible analogues for the ridges beneath the FIS grounded ice.

L29: I do not understand "become spots at the ice shelf base when adjusting the hydrostatic equilibrium"

L32-33: I do not think that this manuscript provides information on the "type of material and structure of the bed upstream of a basal channel". Yes the apparently offline seismic reflection in figures 2&8 is analysed to suggest that it is composed of unconsolidated sediments, but it certainly doesn't reveal the structure of the subglacial bed.

L4: it is interesting to note that the modelled subglacial outflow in figure 1b does not map exactly against the ice shelf channel (i.e. they are offset by ~5 km despite a new high-resolution bed topography of the SFG trunk).

L8: "into the precise"

L9: "typically penetrate"? There are some examples.

L11: I do not understand "..or does the substrate also consist of sediments in which case we can expect recent sedimentation on the seabed?" – even if the bed is hard bedrock you can still have sedimentation on the seabed by (a) subglacial erosion of bedrock; and (b) transport of sediment from upper parts of a glaciers catchment underlain by sediment.

L16: "supported" rather than "backed up"?

L17: "continue" rather than "proceed"

L18: "active/current" subglacial drainage?

L23-24: Support Force actually lies between Academy Glacier and Recovery Glacier. It may also be worth making clear that "northwest" is "grid northwest".

L24: "constrained" rather than "tugged in"?

L30: reference Bedmachine as well/instead of Bedmap2?

L33: I would not describe the survey as a "grid". Refer to figure 1 at the end of line 33.

L2: "Ice surface velocities"?

L11: I do not believe that the DLR acronym has been defined earlier in the manuscript.

L14: Jeofry et al. ESSD, 2018 might be a better reference than Rippin et al.? see https://essd.copernicus.org/articles/10/711/2018/

L14: think you mean 312.5 Hz here? See section 2 of https://agupubs.onlinelibrary.wiley.com/doi/full/10.1029/2018GL077504

L16: "laser surface terrain"?

L27: Paden et al. reference is 2010 in reference list? Please also be specific about which years of OIB data were used (e.g. up to Dec 2018 Antarctic surveys)?

L30: how extensive was the model domain for the hydrological routing presented in 1b? Was it just the domain shown in 1b, or was it more extensive (e.g. entire SFG catchment)?

Figure 1: Please annotate start and end of seismic lines in figure 1 and corresponding seismic profile figure.

L1: as stated previously, the modelled subglacial meltwater outflux location does not correspond that well with the surface channel, despite the new high resolution ice thickness/bed data.

Table 1: Why is it important to have the column "position from GL"? From which part of the profile is this measured? Suggest deleting it.

Pages 7-9

Given the expertise of reviewer 1 I have not assessed the description of the seismic methods in detail.

L16: After "ice shelf thickness." it might be a good idea to refer to figure 1 to provide an example of the artefacts.

L18: "structure of the seafloor" or "morphology of the seafloor"?

L20: Figure 2 should be split into 2a and 2b, rather than TOP and BOTTOM. Therefore, sentence should begin "Figure 2a shows. . .."

L20 (and throughout manuscript): Delete references to along-profile and across-profile, just label the profiles I-V (e.g. "profile I" rather than "along-profile I").

L21: Start "Figure 2b. . ."

L24 (and throughout manuscript): "reflectivity zones" rather than "intervals". These zone should also be clearly marked on figure 2.
L28: call figure 1 after "interferometry"

L30: "uncoupled" – uncoupled from what? Do you mean "floating"?

Figure 2: label 'a' and 'b'. Need to annotate the start and end of the seismic line (see comments on figure 1). Note that where "bed" is annotated in 2a it is not actually the 'bed'. It is the subsea sediments. What is the weak reflection between ~SP46 and ~SP160 (between the labels "ICE" and "BED")? Please define the different reflectivity zones in figure 2 (see comments above). Provide zoomed-in views of all the detailed features described in section 3.2.

L1: "....elongated feature above the flat bed...."?

L3-4: I suggest that this reflection not be described as a "subglacial drainage feature" at this point in the manuscript. Later in the paper, the reflection is interpreted as a "landform" anyway, so it is very confusing. Simply describe it a reflection at this stage, and then only once section 4.3. has been worked through should it be described as a "landform". It would be useful to have the zoom in of the reflection in figure 2 rather than figure 8.

L6 (and throughout manuscript): "anticlines" – this is a specific geological term not normally used in the description of morphology. I recommend "concave cavity" instead.

L8: "ocean cavity thickens" (rather than deepens)?

L10: what is the observation that constrains the sediment thickness to 200 m? I don't see a clear sediment-bedrock reflection at 200 m in Figure 2, so do the authors instead mean "of at least 200 m"?

L10 (and throughout manuscript): what is meant by "transparent"? How can a material be transparent yet stratified and disturbed? Page 12

L15: Call figure 1 after "at the same distance"?

L20: call figure 3 after "have been marked"?

L21: After "the seabed" add "so is not presented here"?

L22: "sequence of stratified sediment" rather than "stratification sequence"? If the authors are trying to avoid interpretation here, then it shouldn't even be "stratification sequence", it should be a geophysical description like "a series of horizontal reflections".

L23: "of stratification below the basal channel" rather than ".. of a stratification sequence in the basal channel". The seabed and sub-bottom sediments are not in the basal channel.

L24: call figure 3a at end of sentence.

L31: "terraced" not "terrace-shaped"

L32 (and throughout manuscript): instead of "as indicated in Figure 4" just have "(Figure 4)". There are a lot of wasted words throughout the manuscript when figures are being called. Authors should make their statement/describe the data etc. and then simply cite the relevant figure at the end of the sentence. See Box 1 of https://aslopubs.onlinelibrary.wiley.com/doi/full/10.1002/lol2.10165 for a better explanation of what I mean. There is also no need to repeat figure caption information in the text (e.g. page 12, line 26 opening line of section 3.4).

Figure 3: what are the reflections above the seabed? It is not clear in the figure what all the complex reflections are. More annotation is required.

Figure 3: Again, authors should not refer to sub ice shelf sediments as "bed". Bed should only be used when referring to grounded ice.

L3-7: This paragraph should refer to figure 4.

L6-7: I do not think that I understand the sentence "The across-profiles.......across-profile III". I have a suspicion, however, that this sentence might actually be quite key to the authors' suggestion that there is a relationship between the ice shelf channels and the sediment beneath it. If I understand it correctly, here they are suggesting that there is a thicker stratified sequence in the parts of the sub-sea sediments directly beneath the ice shelf channel. Is that correct? At the very least, the authors need to annotate the thicker sequence below the ice shelf channels to assist the reader understand exactly what is being described here.

Figure 4: This figure appears to be critical to the argument that the sediments beneath the ice shelf channel were deposited by the ice shelf channel (i.e. there is a spatial coincidence between the channel and a thick sequence of sediments subsea). What is the thickness of the sediment package beneath the ice shelf channel in figure 4a (note that figure subplots need to be labelled throughout the manuscript)? Is it, as implied by the y-axis, 400 m? If so, that is a phenomenal amount of sediment to be deposited 40 km from the grounding line solely from melting beneath an ice shelf channel. The authors should calculate the sedimentation rate for this package of sediment. Is it possible for their sedimentation rate to be valid (e.g. assuming a certain proportion of sediment in the ice, and known melt rates for ice shelf channels as calculated from ApRES). Figure 4 needs to be better described in the text and the key features (e.g. sediment packages) need to be better annotated. I assume that the ages of transit time from grounding line are based on current ice velocity (i.e. figure 1a)?

Figure 5: why not just make this a 3D figure and show all 4 profiles (figure 2b of https://agupubs.onlinelibrary.wiley.com/doi/full/10.1029/2010GL042884 is an example of what I mean)?

L5-12: There is lots of text in this paragraph that is direct repetition of the figure caption

of figure 6.

L5: "We selected ten across ice radar profiles. . . ."?

L6: what is the evidence for the subglacial landform "shaping the channel upstream of the GL"? How does it do this, and what is the process?

L11-12: This sentence needs some expansion to make 100% clear the spatial coincidence between the landform and the ice shelf channel. Reference to figure 1 would help too. I don't understand the phrase ". . .after which the landform become indistinguishable from the bed"? Surely if it is a basal landform it is the bed? It would also be useful to get a better idea and description of the wider bed topography around the landform (e.g. entire bed of SFG) to understand the context. Figure 1b is of little use in this regard – its colour scheme is very uninformative.

Line 15: again, lots of text that should be in the figure caption, or already is.

Figure 6: Indicate very clearly which radargrams are over grounded ice and which ones are over floating ice.

Figure 6: are the authors absolutely sure that radar profile 5 is fully grounded all the time? Could there be tidally-induced grounding line migration?

Section 4.1: It is a little difficult to link section 4.1 with figure 1, as the text in section 4.1 refers constantly to shot points, but these are not apparent on figure 1.

Line 14: "topographically constrained flow"?

Figure 7: it took me quite a while to figure out exactly what this figure was. I suggest that it simplified by removing the bed profile picks. The key point of this figure is to conceptualise the idea of the offline reflection. Is the red semi-circle a "possible drainage feature" or it is a "landform"?

Section 4.3: I recommend not describing the feature being evaluated as either a "subglacial drainage feature" or a "landform" until the authors actually determine which of the two hypotheses are their preferred one. Whilst the unusual (offline?) reflection they describe is referred to throughout as a "subglacial drainage feature" the author then state on page 19 lines 16-17 that they "prefer interpretation 1" which is that the reflection is from the landform. This is a bit of a mess, and suggests that the authors have changed their preference during the writing of the manuscript but not updated all parts of the manuscript. A "landform" is not a "subglacial drainage feature".

Section 4.3: change heading to "Does the seismic data record a subglacial drainage feature or a subglacial landform?" or something along those lines. Section 4.3 evaluates these two hypotheses on the basis of the seismic data and geophysical theory. The section heading should reflect that in some way.

Line 13: provide some additional detail about what is meant by a "separate drainage feature on a hard bed" – In essence this is a Röthlisberger (R-) channel incised into the overlying ice and should be described as such here.

L17-20: This is an important admission here, and one that is entirely inconsistent with the title of the manuscript "Subglacial sediment transport upstream of a basal channel…..". So far, the data don't even unequivocally demonstrate the presence of subglacial sediments.

L14: "floating ice" rather than "uncoupled ice"?

L16 (and throughout manuscript): what do the authors mean by "disturbed"? This needs defined and highlighted/annotated in a figure. Do the authors mean "deformed"

sediments or stratigraphy?

L17-19: apart from the reflection coefficient values and the "disturbed and stratified" stratigraphy, what other lines of evidence for these materials being grounding line deposits do the authors have? This is where reference to glacial geological literature is essential.

L31: "...are positioned on the western side of SFG near its shear margin."?

L32: I disagree that you can track the landform at least 7.7 km. It is not apparent in profile 3 (Figure 6), so it can therefore be tracked for a maximum of 5.2 km (i.e. up to profile 4).

L33: "some degree of consolidation" – so is it unconsolidated sediments, or not?

L1-23: I am totally unconvinced at present by the argument the authors make linking the sub cavity sediment stratigraphy with the ice shelf channels. I see no evidence at all that (a) sediment is being discharged at the grounding line from a subglacial hydrological channel; or (b) that sediment is being deposited directly from basal melt within the ice shelf channel. There seems to be a huge leap of faith being made in this part of the discussion, particularly in L4-5 i.e. "Taking the evidence together we conclude the landform is hosting the transport of sediments that are deposited in the ocean cavity close to the GL" – I see absolutely no evidence presented in this manuscript supporting the transport of sediments. If the authors wish to pursue this angle in a revised manuscript then they will also need to explain the transport mechanism. Is it subglacial deformation advecting sediment to the grounding line? Is it sediment transport by subglacial meltwater? Or, is it englacial sediment transport and then melt out?

L12-14: Here, the authors state "What we do see is more stratification in all across-profiles below the sub-shelf channel than outside of it and that this stratification extends to the eastern side of across-profile III." If this is the case then this is potentially

important, but at present (except perhaps a hint on page 13) this observation is not effectively presented in the current version of the manuscript. This needs strengthened considerably if the authors are to underpin their argument robustly. I remain unconvinced though that their survey layout is extensive enough to permit this statement. I would also like to see an assessment of the implications of thinner ice (and therefore < englacial signal attenuation) over the ice shelf channels – could this lead to higher amplitudes reflections from subsea interfaces beneath the channels compared to the sediments beneath the thicker ice beyond the channels? I also think that the authors need to carefully consider the entire sediment package (i.e. the stratigraphic relationships between the sediment beneath the ice shelf channels and those beyond the channels).

L29: My understanding of reflection coefficient analysis is that it characterises the physical properties of the upper few metres below the interface. As such it is a stretch to state "the eastern side of the landform consists of sediments...". Perhaps make this statement more specific (e.g. "Reflection coefficient analysis indicates that the upper few metres of the landform is unconsolidated sediment...")?

L31: OK, so here, finally in the conclusions section, the authors are specific about the actual process they believe is at play, i.e. "The landform hosts a channelized subglacial drainage which transports sediment downstream". I therefore ask the following questions (1) what is the evidence for subglacial drainage? (2) how does a landform "host" channelized subglacial drainage? (3) where is the evidence for sediment transport?

L32: How do the authors know that the 200 m thick sediment package has any association with the current processes at the grounding line? These sediments could be ancient and have nothing to do with modern-day grounding line processes. The spatial relationship could merely be coincidence.

L18: Since reviewer 1 has sung the praises of Bradley Morrell, I will sing the praises of

Dave Routledge - A brilliant field guide - he's also great!

Final comment: I appreciate that the majority of comments above will be viewed by the authors as perhaps overly negative. However, I do want to emphasise to the authors that I have provided the comments above because I feel that the acquired data are excellent and potentially very important. I would certainly like to see these data and results being published in some way, but I do believe that a stronger more carefully thought-through and coherent argument needs to be developed to place the assertations and findings put forward on a more secure foundation. I do hope that the comments provided above will assist the authors to achieve this.

Dr Neil Ross Newcastle University 3rd July 2020

---

## Author Comment (AC1) · 16 Aug 2020

Dear reviewers,

Many thanks for your input in helping improve this manuscript. My apologies for the delay, it took longer than I had expected.

General comments: We would like to adjust our interpretation: The off-nadir reflections probably come from the subglacial channel connecting to the basal channel. Through interaction with the warmer ocean the subglacial channel increases its when approaching the grounding line. In our answer to reviewer #2 we explain our adjusted interpretation elaborately.

Abstract: Agreed.

P2 L7-8: Agreed.

P3 L12: Corrected.

P3 L16: Agreed. Both the seismic and radar surveys took place in January 2020, the radar survey shortly after the seismic survey.

Figure 1: Agreed.

P6 L3: Agreed.

P6 L4: Indeed, thanks.

P6 L5: Agreed.

- P6 L7 :Well spotted, thanks.
- P6 L10: Yes it does make the text more readable. Agreed.

P7 L9: Thanks we will.

P7 L11: Good suggestion.

P7 L14 : Agreed.

P7 L20: Agreed.

Table2: Agreed.

- P9 L15: Correct, based on the center frequency.
- P9 L19: Very good point, indeed we mean uncertainty.

Section 3.1: Agreed, it is not an artefact. We'll use "Seabed-depth conversion".
Figure 2: Yes we agree and will add the schematic versions. Regarding the subglacial feature, I think the raw shots give the best indication we are dealing with reflections and will add them here. Lastly, I'm glad you appreciate the lay-out of figure 2.

Figure 2 caption: Agreed.

P11 L3: Yes we agree, we'd like to call this the "subglacial feature" .

P11 L7-8: Yes we think so, the loss must be greater at the gradual ice-seawater contact. We point this out in the discussion (page 20 L6-9) as a general comment, not restricted to interval 2. However in interval 2 the seabed contact occasionally switches polarity, which suggest small magnitudes and we get pretty high amplitudes from deeper down the sedimentary sequence with chaotic reflections.

P12 L17: Correct this happened unintentionally so we will use your suggestion, thanks.

P12 L17-19: We prefer to take it out. The figure is easily misunderstood.

P12 L20-21: The double ice-sea contacts and seabed reflections are caused by different pathways and thus different reflection areas of the seabed. Correct depth converting is actually impossible, what horizon do you pick for depth conversion? Either choice of ice-seawater contact (channel crest or base) will only partly convert the reflections to the correct depth.

Figure 4: Thank you, detonating cord at firn works very well.

Page 15 L15-18: Agreed.

Figure 6: Agreed.

Section 4.3: Agreed.

P18 L11-12: I will add the raw shots, showing the feature (Fig1:profileI\_subglacial\_feature\_SPs.png). The feature is visible over 1200 m. That is a long distance for a diffraction. Especially SP 15, where we see a reflection
splitting off the subglacial feature, shows they are probably reflections. If they had been diffractions I would expect to see them cross each other. Please let me have your judgement again with this extra image. Thanks.

P20 L4: Indeed, we mean magnitude and will add this.

P20 L16: Indeed the term disturbed is not well chosen, we'd like to use chaotic so we refer to a "sedimentary sequence with chaotic reflections". In our answer to reviewer #2 we answer this more elaborately.

P22 L5-6: We'll use planar. Thanks.

P22 L19: Yes now actually Daniel (Steinhage) worked with Bradley and I was shown around by Dave Routledge. We met each other at the shelf of Support Force Glacier and then did this survey together. It worked like clockwork.

Supplement: Thank you, this is highly appreciated.

Best regards, Coen Hofstede

---

## Author Comment (AC2) · 17 Aug 2020

My apologies for the delay, it took longer than I had expected. Please note these are comments on reviewer #2

General comments 1:

Thank you for making this point. We overlooked this and will back this up with literature and a more focused interpretation. We'd like to stress that we make an interpretation of the radar and seismic profiles so at best evidence is circumstantial. However, we do

believe the interpretation we provide is the best explanation of what we observe in the seismic and radar profiles. This is also supported by the evaluation of Adam Booth, reviewer #1, one of the seismic experts in glaciology.

In our answer we'll use the following terminology:

Grounded ice:

- Subglacial channel: a feature between the ice and the bed, probably water filled. Needs ice to be visible.

- Landform: a geomorphological feature of the bed – would be visible without the ice

Ice shelf:

- Surface channel: Meandering narrow long channel at the surface of an ice shelf

- Basal channel: The sub-ice shelf channel causing the surface channel through hydrostatic adjustment.

The seismic survey concentrates at a surface channel caused by a basal channel at the ice shelf of Support Force Glacier. The basal channel is formed upstream by a subglacial channel we see in radar profiles. At the grounded ice we can track the subglacial channel at radar profiles 3, 4 and 5. At profile 3, 7.1 km upstream from the grounding line, the subglacial channel is hardly distinguishable from the bed after which its height increases at profile 4, 4.4 km upstream from the grounding line, to approximately 100 m above the surrounding bed. At profile 5,1.8 km upstream from the grounding line, the top of the channel increased to approximately 250 m above the surrounding bed. Profile 6 lies at the grounding line: the western part has passed the grounding line, the eastern part has not. The basal channel, an extension at the ice shelf of the subglacial channel, now reached a height of approximately 300 m above the surrounding base of the ice shelf.

This is where we'd like to adjust our interpretation: Considering the comment of reviewer #1 that the refection coefficient of the off-nadir reflections is tricky (we don't think they are diffractions) and we might over interpret the data, we will only use its polarity which indicates the presence of water. The radar profiles show the subglacial channel increases its height from approximately 0 m to 300 m over a length of 7.1 km approaching the grounding line. This would place a landform within 7.1 km upstream of the grounding line which we think is unlikely. Summarizing, if we leave out the value of the reflection coefficient we see no evidence the off-nadir reflections in seismic profile I between radar profiles 5 and 6, are caused by a landform.

We interpret these reflections to come from the subglacial channel we see in the radar profiles 3, 4, 5 and 6. The increase in size, when approaching the grounding line, is likely caused by the ocean is interacting with the subglacial channel due to tidal motion thereby increasing its size due to melting of the channel walls as suggested by Drews et al. (2017), Horgan et al. (2013) and modelled by Walker et al. (2013). The radar profiles 3, 4, 5 and 6 show the subglacial channel interacting with the warm ocean. Once passed the grounding line this wide opening of the subglacial channel adjusts to hydrostatic equilibrium and forms the basal and surface channel in which the subglacial drainage water incises.

We plan to adjust our interpretation accordingly: At the grounded ice of Support Force Glacier radar profiles 3, 4, 5 and 6 show a subglacial channel connecting a basal at the grounding line. Approaching the grounding line the subglacial channel increases its size to 300 m height at the grounding line, which we attribute to ocean interaction. This setting is similar to the subglacial estuary described by Horgan et al. (2013). Because the subglacial channel connects to the only basal channel at the western side of the ice shelf, and because we have a large subglacial drainage influx modeled at the western side of the ice shelf, we interpret the subglacial channel to be a subglacial drainage channel.

The grounded part of profile I consists of a sediment layer judging by its reflectivity becoming more consolidated closer to the grounding line. So the drainage channel

probably travels over a layer of subglacial sediments with varying consolidation. The exact nature of the subglacial drainage system we do not know but the radar and seismic profiles do suggest channelized flow close to the grounding line. Possibly we are dealing with a channel that, upstream and outside the survey area, is coupled to a surrounding distributed system as described by Hewitt (2011). Close to the grounding line channelized flow is favorable which corresponds to our observations.

General comments 2:

To summarize our findings: We have a modelled large influx of freshwater on the western side of the shelf.

From the airborne radar data of the shelf we know the ice shelf has only one basal channel at the western side. That must be the place where the subglacial drainage channel enters the ocean.

There is a noble gas sample downstream of Support Force Glacier suggesting a freshwater influx of terrestrial origin coming from Support Force Glacier.

Along profile I (along-flow, 1.5 km east of the basal channel) shows an approximately 200 m thick sedimentary sequence close to the grounding line of different character then the seabed further downstream part of the ocean cavity. The sedimentary sequence is less consolidated and has chaotic reflections with high amplitudes. Across profile III, crossing this sedimentary sequence with chaotic reflections, shows this sequence is only present under the sub-shelf channel. Both on the far east and west side of profile III there is hardly any structure in the seabed except right under the channel. This sedimentation most likely has been transported by the subglacial channel.

Based on profile I and III we interpret the sedimentation to be point sourced and fan shaped, possibly a grounding line fan (Powell, 1990) or an ice-proximal fan (Batchelor and Dowdeswell, 2015). This explains the chaotic reflections (we referred to as disturbed), with high amplitudes in this sedimentary sequence and this material being

softer as the further downstream part of the sea bed.

We realize there are concerns here as the fan has formed under an ice shelf of Support Force Glacier without surface melt, a characteristic of fans (Powell and Alley, 1997). But we do have evidence for channelized flow at the grounding line, a noble gas sample suggesting freshwater observation influx of terrestrial origin likely (Huhn et al. 2018) and a significant (190 x 106 m3 a-1) modelled channelized freshwater influx at one place on the west side confirmed by the presence of a single basal channel on the western side. We also have an unusual ocean cavity with a steeply descending seabed and, as argued in our paper, a stable grounding line. These are typical conditions for the formation of a fan at the grounding line (Powell 1990, Powell and Alley 1997, Batchelor and Dowdeswell 2015) . We will emphasize this in the text and update figure 4 with a schematic lay out as in figure 2 where we identify the sedimentary with chaotic reflections.

Can we proof all this and can we say how old this sedimentation process is? Not without sea bed samples of the sedimentary with chaotic reflections or an additional seismic across-flow profile passing the subglacial channel. Do we think this interpretation is likely and sound? Yes we do if we look at the glaciological setting; a grounding line environment where a subglacial drainage channel enters the ocean cavity with a descending seabed and seismic profiles show a sedimentary with chaotic reflections right under the basal channel.

General comments 3:

In our reaction to general comments 1 we explain our adjusted interpretation: the off-nadir reflections are probably caused by the enlarged opening of the subglacial channel, not the landform. In our reaction to general comments 2 we explain why, based on seismic profile I and III we interpret the sedimentary sequence with chaotic reflection to enter the ocean cavity through the subglacial channel.

We agree we should make a better case here. We've set out our reasons as to why

we think we can connect the sedimentary sequence with chaotic reflections we see in seismic profile I and III to the subglacial channelized flow and why we think this sequence probably resembles a fan.

Conceptual model: Is this like a picture explaining the model? It should be possible but it is quite some work, There are figures showing the formation of fans at grounding line like in Powell (1990). The difference in our case is that there will be a shelf at the grounding line instead of a cliff. But if you feel the paper needs it, we can provide it.

Specific comments:

Title: Updated.

Title evidence: Indeed sediment transport is an interpretation mainly based on the structure and reflectivity of the seismic sections presented as such we suggest: "Likely subglacial sediment..."

Addresses: Updated.

Abstract L1: Agreed, the surface channel at Support Force Glacier starts as a meandering surface channel at the grounding line, is not a flow stripe.

L5: Corrected. Floating part should be ice shelf.

L8: Agreed.

L10:It is an interpretation we give in our reply to the general comments. We will adjust the discussion text accordingly and will explain why we interpret this sequence as grounding line deposits.

L10: This will be removed as we interpret this feature no longer as a landform.

L15: Indeed the channel is 4 km east of the shear margin. We will remove this.

L4: We'll remove the association with flow stripes: "They are often detected with satellite imagery like MODIS. . ."

L25-27: We disagree. Jeofry et al.2018 suggest a combination of a landform incising the base of the ice which when becoming afloat will cause a surface channel and basal channel. Quote: "we propose that the bedforms are dictating the position and form of the U channels." Which is also why they checked the dimensions of landforms that are indeed at completely different locations which we also state in L26, 27. As the landform also organizes the drainage pathway, quote:"the water incises upward into the corrugation peak" also because fresh water will want to move upward will assemble in the by bedrock formed corrugation peak.

L29: The surface trough of the shear margin (that has a surface depression) induces a basal channel due to hydrostatic adjustment once it passes the grounding line. Once afloat, the surface trough is shallower while adjusting but then deepens again as a warm water plume thins the base of the ice in the channel.

L32-33: Indeed, we state that this observation is often missing, nothing more.

L4: This is correct, the modeled drainage pathway is offset by 4 km from the basal channel. This model is coarse, it has a resolution of 1 km and does not take the physical nature of the bed into account that may steer the pathway somewhat differently. Although we did use the topography derived from the airborne radar data, the surrounding is of course still BEDMAP2. So the model is an indication of where one may expect a subglacial drainage system.

L8:Agreed.

L11: Thanks for making this point, we will rephrase "or does the substrate also consist of sediments."

L16: Agreed.

L17: Agreed.

L18: Agreed.

L23-24: Thanks, corrected.

L24: Agreed.

L30: Just BEDMAP 2.

L30: Agreed.

L2 : Correct, thanks.

L11: We will correct this.

L14 Jeofry et al: Good suggestion, thanks.

L14: Correct.

L16: Correct .

L27: Thanks for bringing it to our attention. The reference should be: Paden, J., Li, J., Leuschen, C., Rodriguez-Morales, F., and Hale., R.: 2010, updated 2018. IceBridge MCoRDS L2 Ice Thickness, Version 1. Antarctica., Boulder, Colorado USA. NASA National Snow and Ice Data Center Distributed Active Archive Center.,https://doi.org/10.5067/GDQ0CUCVTE2Q, 2019.

L30: The model domain for the routing was the entire hydrological catchment of SFG. I attached a small figure, which we could put into the answer or appendix.

Figure 1:We'd like to use an arrow head at each line.

L1: As explained (Page 3, L4) we use a model with its shortcomings. The main result

[Figure]

is we can expect water drainage on the western side of the shelf. Keeping in mind that the resolution of the model is 1 km, and does not take the physical properties of the bed into account, we find 4 km acceptable.

Table1: We will remove this.

L16: Reviewer #1 pointed out that the phrase artefact is not accurate. We'd like to change the title to "seabed depth conversion" as he suggests.

L18: The morphology is a better phrase.

L20: Agreed.

L24: They are marked with double headed arrows on top of the figure: interval 1, 2, 3 and 4.

L28: Agreed.

L30: Yes we mean floating.

Figure 2: labeling agreed. The shot numbering gives the shooting directions, but we can add this if you feel this is not clear.

Note: Is subsea bed acceptable? If we talk about sediments we are interpreting.

Weak reflections: The are probably side reflections of the ice-sea contact, the polarity is reversed just as the identified sea-bed contact which is why we think they resemble ice-seawater contacts. The shelf base here has a lot of topography.

Reflectivity zones: As mentioned we have defined the reflectivity zones. Please let me know if you find this ok. We can provide zooms of key features.

L1: Based on shot spacing and two-way travel time the dimensions are 1200 m long and the feature appears to be 50 m higher than the surrounding bed if it was nadir, hence we called it elongated although of course we do not know the across-flow dimension.

L3-4: We propose to call this "subglacial feature". We will provide a zoom of this subglacial feature in our response to reviewer#1 as we think we are dealing with reflections. As mentioned this part of our interpretation we want to change: We interpret this subglacial feature still as off-nadir reflections but no longer as a landform but as the top of the subglacial channel that in this area so close to the grounding line likely interacts with the ocean. This interaction with the ocean probably caused a rapid increase in height of the subglacial channel.

As reviewer #1 pointed out, a quantitative analysis its reflectivity is tricky as we have a complex subglacial structure off-nadir. To avoid over-interpretation we will not use the calculated reflection coefficient but its polarity.

L6: Agreed, concave cavity is a better term.

L8: Agreed.

L10 200m: It is as you observe, there is not a clear last sediment-bedrock reflection but the chaotic reflections fade out with increasing depth. Hence approximately 200 m.

L10 transparent: What we mean by transparent is that the seismic signal penetrates deep in the formation with little loss of amplitude.

We will use the phrase sedimentary sequence with chaotic reflections with high amplitudes as mentioned in our reaction to general comments 2.

L15 Agreed.

L20: As this a complicated profile with two ice-sea contacts and two sea-seabed contacts, we'd like to take this figure out of the paper as reviewer 1 suggests. We hardly use it in our interpretation and the figure is complicated to explain.

L21: Yes the seabed is present here as we have the seafloor returns twice (two different ray paths: path 1 is along crest of the channel, path 2 is along base of the ice next to the channel and they likely have the same seabed depth. Converting the time migrated section to depth is not really possible as we must choose one of these two ray paths to convert to depth but then we automatically misplace the reflections of the other ray path.

L22: We will remove Figure 3 as profile II is difficult to interpret and probably causes misunderstandings. As reviewer #1 pointed out, profile II hardly contributes to the interpretation.

L23: Will be removed.

L24: Will be removed.

L31: Thanks.

L32: Thanks for pointing this out.

Figure 3: There are two seabed reflections present due to different travel paths. See our reply at L21.

We will remove Figure 3

Figure 3 bed: Thanks, we will

L3-7 Agreed

L6-7: Yes that is what we claim, right under the basal channel, profile III shows thicker stratification (roughly from SP 3 to SP 24) under the basal channel then outside the basal channel. We plan to add schematic images (recommendation of reviewer 1) of

profiles 3,4 and 5 as in Figure 2 marking the stratification areas.

Figure 4: Indeed I see the confusion. Profile V (as profile IV) has multiples causing apparent stratification. This is clearer to spot in the time migrated profiles. So no, there is not 400 m of stratification at profile V (Figure 4a), I come to no more than 100m. We will provide a schematic picture with our interpretation. The focus of the paper lies on the sedimentary sequence with chaotic reflections that profile III crosses.

Figure 5: Thanks for the suggestion. What we like to provide is use both radar and seismic profile to show the development of the subglacial channel (grounded) and how this continues as a basal channel under the ice shelf. The present figure actually consists of 3 profiles, profile 4 is used twice. Reason why we displayed them like this is to get a good handle on where melt/widening of the basal channel takes place.

L5-12:We will clean this up.

L5: Agreed.

L6: We withdraw this interpretation.

L11-12: We will adjust our interpretation as we state in our reaction to general comments 1 and remove the concept landform. Looking at the radar profiles 3, 4, 5 and 6 we see that the subglacial channel we see at the grounded ice, increases in size as it approaches the grounding line. So there is no landform at the grounded ice, just a subglacial channel that increases its size due to interaction with the ocean.

L 15: Agreed.

Figure 6: Agreed that should be made clearer. Profile 6 lies at the grounding line.
Figure 6 profile 5: We have no indication profile 5 is susceptible to grounding line migration. Profile 5 crosses seismic profile I at SP 5 where we have a positive basal reflection indicating consolidated material. To us that means the ice is grounded here. If ocean water would have reached this far it would have influenced the reflectivity. We do have an indication the MOA grounding line, crossing seismic profile I at SP 51, is not correct. Seismic profile I clearly shows ocean water being present upstream of the MOA grounding line down to SP 26. Are we absolutely sure profile 5 is fully grounded all the time? No but it is very likely.

Section 4.1: We will be clearer here. We wish to refer to figure 2, profile I here, and will add this in the text. The interferometric grounding line crosses profile I at SP 23 but this can't be chosen that precise, . The polarity switch at profile I lies at SP 26, so 150 m downstream of SP 23. This deviation may be caused by the unprecise choice of the grounding line here.

L14: Correct.

Figure 7: We think the concept of Figure 7 is still not clear.

The figure should show that off-nadir reflections of the landform (represented by the radar profiles 5 and 6 and we now interpret as the subglacial channel) arrive at the same time as if there had been a 50 m high channel at nadir (represented by the red semi-circle). As reviewer #1 points out, the weakness of the reflections shown in figure 8 (a zoom of profile I, figure 2) already suggest these reflections (or diffractions as reviewer 1 points out) are off-nadir.

Section 4.3: Indeed, we adjusted our interpretation as described in our reaction to general comments 1 and will adjust the text accordingly.

Section 4.3: We will restructure this according to our interpretation: The reflections are off-nadir and represent the subglacial channel. The channel opening is enlarged here

due to interaction with the ocean. This interaction between ocean and a subglacial channel is described by Horgan et al. (2013).

L13: Indeed if the reflections are at nadir it would seem like an R-channel and that is represented by the red semi-circle. That this is most likely not the case is because there is only one basal channel visible at the western side of the ice shelf and we argue that this is where the subglacial channel enters the ocean cavity which is on the western side so off-nadir of profile I. Had the reflections been nadir, the R-channel would have entered the ice shelf elsewhere but we see no evidence of another basal channel in the radar data. That is our main argument as to why we think the reflections are off-nadir and are caused by the subglacial channel.

L17-20: Correct, it is an interpretation.

L14: Correct.

L16: We propose chaotic reflections with high amplitudes, as mentioned in our reaction to general comments 2.

L17-19: We are presenting an interpretation. Seismic profile I and seismic profile III most likely show the presence of a grounding line fan.

L31: Agreed

L32: If you follow the same flow line along profiles 3, 4, 5 and 6 (marked on the long profiles with an arrow and radar trace number it is quite obvious. We also have 38km long profiles that make a clearer case for this observation which we will provide as Figure 2.

L33: As we pointed out in the our reaction to general comments 1 we will withdraw the quantitative analysis of the reflectivity. We will just use the polarity of the off-nadir

reflections.

L1-23: There is clear evidence of subglacial drainage at the western side namely the basal channel itself which matches a modelled subglacial drainage pathway with a large water flux. The radar profiles 3, 4, 5 and 6 indicate the presence of a subglacial channel matching the location at the grounding line of the basal channel. The increase in height of the subglacial channel seen on profiles 4, 5 and 6, close to the grounding line can very well be explained by interaction with the ocean. This is what one would expect of channelized flow close the grounding line and has been suggested by Horgan et al. (2013) and Drews et al. (2017) and modelled by Walker et al. (2013) and Hewitt (2011). What we can't proof is that this channel is carrying sediments but it is likely that at the end of an ice stream the subglacial channelized drainage system carries sediments. We do have the observation in seismic profiles I and III, of a sedimentary sequence with chaotic reflections close to the grounding line (profile I) and the presence of this package only under the basal channel (profile III), exactly where one would expect sedimentation to take place if the subglacial channel would be carrying sediments.

L12-14: Profile III shows thick sedimentation only under the basal channel consisting of several levels and extending eastward. We agree we should emphasize this observation and it's interpretation more. This is what links the sedimentation to a grounding line fan where the subglacial channel enters the ocean cavity and forms the basal channel by adjusting to the hydrostatic equilibrium. Profile IV and V have also show sedimentation but are tricky as multiples occur between stronger reflections. See my reaction to your comments at Figure 4. These profiles also cross different formations that are beyond the focus of the paper. When calculating a reflection coefficient, the attenuation in ice and seawater over the entire travel path are taken into account as is pointed out in chapter 2.6, equation 2. As such reflectivity is compensated for the attenuation.

[Figure]

L29: The reflection coefficient characterizes the interface between two media but if there is a layered sequence the reflection coefficient can be influenced by interference. We will just stick to the polarity of the off-nadir reflections.

L31: Evidence for subglacial drainage I pointed out answering your comments at page 21, L1-23 The subglacial feature is most likely the subglacial channel interacting with the ocean as pointed out in the our reaction to general comments 1. Profiles I and III are evidence of a grounding line fan under the basal channel.

L32: As mentioned quite extensively in our reaction to general comments "we do have evidence for channelized flow at the grounding line, a noble gas sample suggesting freshwater observation influx of terrestrial origin likely (Huhn et al. 2018) and a significant (190 x 106 m3 a-1) modelled channelized freshwater influx at one place on the west side confirmed by the presence of a single basal channel on the western side. Seismic profile I and III suggest the sedimentary sequence with chaotic reflections is point sourced and fan shaped, possibly it is an ice-proximal fan (Batchelor and Dowdeswell, 2015). This explains the chaotic reflections and this material being softer as the further downstream part of the sea bed. We also have an unusual ocean cavity with a steeply descending seabed and, as argued in our paper, a stable grounding line. These are typical conditions for the formation of a fan at the grounding line (Powell 1990, Powell and Alley 1997, Batchelor and Dowdeswell 2015) ." Lastly we do not provide hard evidence but an interpretation.

L18: Indeed it was a smooth operation.

Final comment: Your comments are highly appreciated. They force us to built up our case better which improves the manuscript. So thank you.

Coen Hofstede, August 15, 2020

References:

Batchelor, C., & Dowdeswell, J., 2015. Ice-sheet grounding-zone wedges (GZWs) on high-latitude continental margins. Marine Geology, 363 65-92. https://doi.org/10.1016/j.margeo.2015.02.001

Drews, R., Pattyn, F., Hewitt, I. et al. Actively evolving subglacial conduits and eskers initiate ice shelf channels at an Antarctic grounding line. Nat Commun 8, 15228 (2017). https://doi.org/10.1038/ncomms15228

Hewitt, I. (2011). Modelling distributed and channelized subglacial drainage: The spacing of channels. Journal of Glaciology, 57(202), 302-314. doi:10.3189/002214311796405951

Horgan, H. J., Alley, R. B., Christianson, K., Jacobel, R. W., Anandakrishnan, S., Muto, A., Beem, L. H., & Siegfried, M. R. (2013). Estuaries beneath ice sheets. Geology, 41(11), 1159-1162. https://doi.org/10.1130/G34654.1

Huhn, O., Hattermann, T., Davis, P. E. D., Dunker, E., Hellmer, H. H., Nicholls, K. W., Østerhus, S., Rhein, M., SchroÌĹder, M., and SuÌĹltenfuß, J. , 2018. Basal Melt and Freezing Rates From First Noble Gas Samples Beneath an Ice Shelf, Geophysical Research Letters, 45, 8455–8461, https://doi.org/10.1029/2018GL079706, https://agupubs.onlinelibrary.wiley.com/doi/abs/10.1029/2018GL079706.

Powell, R.D., 1990. Processes at glacial grounding-line fans and their growth to ice-contact deltas. In: Dowdeswell, J.A., Scourse, J.D. (Eds.), Glacimarine Environments: Processes and Sediments. Geological Society of London Special Publication 53, pp. 53–73

Powell, R.D., Alley, R.B., 1997. Grounding-line systems: processes, glaciological inferences and the stratigraphic record. In: Barker, P.F., Cooper, A.C. (Eds.), Geology and Seismic Stratigraphy of the Antarctic Margin II. Antarctic Research Series 71. American Geo- physical Union, Washington, DC, pp. 169–187.

Walker, R.T., Parizek, B.R., Alley, R.B., Ananda- krishnan, S., Riverman, K.L., and

Christianson, K., 2013, Ice-shelf tidal flexure and subglacial pressure variations: Earth and Planetary Sci- ence Letters, v. 361, p. 422–428, doi:10.1016/j .epsl.2012.11.008.

[Figure]

[Figure]

**Fig. 1.**

[Figure]

**Fig. 2.**

---

## Author Response (AR1)

**RC1:**

Dear authors,

I really enjoyed reading your manuscript - I'm very sorry that it has taken me so long to review it.

*Dear reviewers,*

*Many thanks for your input in helping improve this manuscript. My apologies for the delay, it took longer than I had expected. My answers to the points you raise are written in italic format.*

*Coen Hofstede*

*General comments:*

*We would like to adjust our interpretation: The off-nadir reflections probably come from the subglacial channel connecting to the basal channel. Through interaction with the warmer ocean the subglacial channel increases its when approaching the grounding line. In our answer to reviewer #2 we explain our adjusted interpretation elaborately.*

General comments:
The paper explores a little-studied feature in the base of Antarctic ice shelves, but one that is nonetheless important for ice shelf stability and sub ice-shelf circulation. The study also indicates subglacial sediment transport. The data acquisition and processing is of a high standard and the imaging is unambiguous. Most of the comments in the attachment are suggestions for grammatical corrections or improvements for readability; some of the paragraphs seem a little rushed, and some sentences aren't as easily understood as I think they might be. I also have some more substantive requests for clarification of a few points, as mentioned below, but I don't think that these are major issues. Overall, I think that the paper is a nice contribution to the understanding of ice shelves and their dynamics.

Best regards, Adam Booth Specific comments:

Abstract - I found this a little qualitative, and maybe some hints at the dimensions of the subglacial channel could be useful?

*Abstract: Agreed*

P2 L7-8 - similar to my comment in the abstract, maybe some quantitative detail about the geometry of surface channels could be useful?

*P2 L7-8: Agreed*

P3 L12 - 'active source' seismic

*P3 L12: Corrected*

P3 L16 - if its important to mention the date of the radar acquisitions, do so for the seismic too; I'm not sure you say when the data were acquired at any point in the manuscript.

*P3 L16: Agreed. Both the seismic and radar surveys took place in January 2020, the radar survey shortly after the seismic survey.*

Figure 1. Add a distance scale for the arrow lengths in (a). I also wonder if the figure would benefit from some textual geographic labels (e.g., the landmarks and features mentioned in the intro?).

*Figure 1: Agreed*

Section 2.5 - again, I'd mention the acquisition date here.

*Section 2.5: Agreed*

P6 L3 - define p as compressional (equally, S as shear when it comes up later). Additionally, you occasionally swap between "P wave" and "p wave". Be consistent.

*P6 L3: Agreed.*

P6 L4 - your sample rate is actually a sample interval.

*P6 L4: Indeed, thanks*

P6 L5 - consider putting the manufacturer of the GEODE system... something like "four Geometrics GEODE units"?

*P6 L5: Agreed*

P6 L7 - should II be III, given the information in Table 1?

*P6 L7 :Well spotted, thanks*

P6 L10 - I think your description of the acquisition (particularly for the long offset gathers, but maybe also for the profiles) might benefit from a schematic diagram of the survey.

*P6 L10: Although it does make the text more readable we refrained from this as they hardly play a role in the results and discussion.*

P7 L9 - define R(theta) at the start of this paragraph, otherwise Reflectivity and the symbol in Equation (2) is undefined.

*P7 L9: Thanks we will*

P7 L11 - it's unclear how Vp, Vs and rho relate to the primary reflectivity, in the way that you have described it here. I would consider splitting this sentence, explaining how reflectivity is defined by contrasts in these quantities, and then introducing Equation (2).

*P7 L11: You're right, we've built up this part differently now.*

P7 L14 - just give the Section number explicitly, rather than the cumbersome "determination of A0 subsection".

*P7 L14 : Agreed*

P7 L20 - give references for the parameter ranges you use in Table 2, to provide you with reflectivity ranges.

*P7 L20: Agreed*

Table 2 - explicitly state which material would have subscript 1 and 2 (i.e., which is above and below the interface). You might also consider defining the impedances as well as the Reflection coefficients?

*Table2: Agreed*

P9 L15 - the abrupt transition is, of course, only abrupt on the wavelength scale of your wavelet; maybe simply adding "at the XX m scale of vertical resolution" in here?

*P9 L15: Correct, based on the center frequency*

P9 L19 - here, and throughout, I think you're mis-using the term 'accuracy'. If something has 19% accuracy then it is very poor indeed! Do you mean 19% uncertainty, or "accuracy better than 81%", or suchlike? If I'm correct, that you're mis-using this term, make sure that other instances of accuracy are checked too. In this specific case, it's also not clear to me how the numbers above end up giving this accuracy.

*P9 L19 Thank you, indeed we mean uncertainty*

Section 3.1, header - I'm not sure that these are 'artefacts' as such, which I'd consider more to be residual effects of processing (e.g., migration smiles). I'd suggest that this subsection is retitled "Seabed depth conversion"?

*Section 3.1: Agreed, it is not an artefact. We'll use "Seabed-depth conversion".*

Figure 2 - this is a nice figure, but I'd suggest that the other figures in the section are given the same interpretation panels - it's unclear why you'd only provide it for this one. It might also be good to show an enlarged section of the subglacial feature; I know it features in other figures, but I really couldn't see it here.

*Figure 2: Yes we agree and will add the schematic versions. Regarding the subglacial feature, I think the raw shots give the best indication we are dealing with reflections and will add them here. Lastly, I'm glad you appreciate the lay-out of figure 2*

Figure 2 caption - "switches from positive to negative" - how do you define what is positive and negative polarity? You might just be better saying "changes polarity".

*Figure 2 caption: Agreed but now we define positive and negative polarity in the text*

P11 L3 - it seems a little premature to be referring to this feature as a 'drainage' feature. It's only in the Discussion where you start to present the evidence for this, based on previous work. At the moment, it is a subglacial feature, but it's impossible to know it's a 'drainage' feature from the seismic results alone. I'd suggest that you remove 'drainage' at this point in the manuscript.

*P11 L3: Yes we agree, we'd like to call this the"subglacial feature"*

P11 L7-8 - is the gradational, rather than abrupt, transition the reason why you see the deviation to smaller-magnitude reflectivity? Is it worth making this comment explicitly?

*P11 L7-8: Yes we think so, the loss must be greater at the gradual ice-seawater contact. We point this out in the discussion (page 20 L6-9) as a general comment, not restricted to interval 2. However in interval 2 the seabed contact occasionally switches polarity, which suggest small magnitudes and we get pretty high amplitudes from deeper down the sedimentary sequence with chaotic reflections.*

P12 L17 - The sentence "Consequently..." makes it sound like you did this deliberately, whereas I don't think you did at all! I'd rephrase this as "Consequently, along-profile II samples the west flank of the channel rather than its crest, and therefore complicates the recorded seismic response."

*P12 L17: Correct this happened unintentionally but we'll take the figure out, thanks.*

P12 L17-19 - It took me quite some time to recognise what was going on with the appearance of the 'double bed' in Profile II and Figure 3. I think you need to explain the geometry more clearly, and explain that you have these two laterally-offset reflectors within a Fresnel zone of each other. I also think it would be helped if you presented Profiles III, IV and V first - they don't have to come in numerical order. That said, I do wonder if Profile II adds much to the interpretation - you don't really refer to it later in the manuscript, and it's clearly not acquired in the most ideal location (not that I blame you, of course, it happens!!). Maybe it should be relegated to supporting material?

*P12 L17-19: We prefer to take it out. The figure is easily misunderstood*

P12 L20-21 - Why would depth conversion obscure the sea bed?

*P12 L20-21: The double ice-sea contacts and seabed reflections are caused by different pathways and thus different reflection areas of the seabed. Correct depth converting is actually impossible, what horizon do you pick for depth conversion? Either choice of ice-seawater contact (channel crest or base) will only partly convert the reflections to the correct depth.*

Figure 4 - these seismic images are lovely :)

*Figure 4: Thank you, detonating cord at firn works very well.*

P15 L17-18 - with the likely complex pattern of reflectivity close to the uncoupling, I'm not sure you can say that the polarity 'confirms' the presence of water - but it might certainly support or imply it.

*Page 15 L15-18: Agreed*

Figure 6 - it might be good to include a refresher of the location map?

*Figure 6: Sorry we refer to figure 1 d for the numbering. Hope that is acceptable.*

Section 4.3 - again, I'm not sure that 'drainage' is yet appropriate in this section header.

*Section 4.3: Agreed*

P18 L11-12 - Given that the landform likely represents a diffracting point rather than a specular reflection, I'm not sure that reflectivity calculations hold. I agree with your geometric arguments and think that you do a good set of analyses here, and I think that the reflectivity argument is in any case superfluous. I'm not sure what the reflection coefficient equation would be for this; I think you can likely speculate that the amplitude appears weaker than surrounding reflections, but the quantitative assessment might be an over-interpretation. (this comes back on P19 L17-18).

*P18 L11-12: I will add the raw shots, showing the feature. The feature is visible over 1200 m. That is a long distance for a diffraction. Especially SP 15, where we see a reflection splitting off the subglacial feature, shows they are probably reflections. If they had been diffractions I would expect to see them cross each other. Please let me have your judgement again with this extra image. Thanks*

P20 L4 - define 'trend'. Do you mean the magnitude? As in, you're interpreting based on indicative reflectivities rather than a fully-quantitative asessment?

*P20 L4: Indeed, we mean magnitude and will add this.*

P20 L16 - I wonder if the terms 'disturbed' and 'undisturbed' imply a process rather than a geometry? As in, the implication that the sediment has been disturbed by something (e.g., ocean currents). Of course, this might be the case, but as an indicator of simple geometries then I think that 'stratified' and 'unstratified' of 'homogeneous' might be less weighted?

*P20 L16: Indeed the term disturbed is not well chosen, we'd like to use chaotic so we refer to a "sedimentary sequence with chaotic reflections". In our answer to reviewer #2 we answer this more elaborately.*

P22 L5-6 - I think flat and horizontal might be the same thing? The difference here might be in the terms 'planar' and horizontal.

*P22 L5-6: We'll use flat. Thanks*

P22 L19 - I agree with you, Bradley Morrell is great! Technical issues:

*P22 L19: Yes now actually Daniel (Steinhage) worked with Bradley and I was shown around by Dave Routledge. We met each other at the shelf of Support Force Glacier and then did this survey together. It worked like clockwork.*

There are many grammatical issues which I have flagged up in the attached manuscript. These are flagged up, and suggestions made for alternative wording; all of the comments above are also included.

Please also note the supplement to this comment: https://tc.copernicus.org/preprints/tc-2020-54/tc-2020-54-RC1-supplement.pdf

*Supplement: Thank you, this is highly appreciated.*

**RC2:**

**Neil Ross (Referee)**

neil.ross@ncl.ac.uk

This is a well reported high-resolution seismic reflection survey targeting the form and physical properties of an ice shelf channel and the sub-seafloor sediments beneath it at Support Force Glacier (SFG), East Antarctica. The seismic analysis is supported by some airborne ice-penetrating radar data. The methods are sound and detailed, the description and presentation of the data is reasonably good, and the science is high-quality and of potential interest to the readership of TC. The data are hard won field geophysical data from a remote part of the Filchner Ice Shelf, Antarctica and certainly deserve to be published in some form. I do, however, have some serious concerns about the way the data are 'pitched' and argued in the current version of the manuscript. Specifically, I am very unconvinced by the association between the ice shelf channel, the "subglacial landform", and the sub-seafloor sedimentary structures, and the argument for sediment transport subsequently developed.

General comments 1. The manuscript does not engage at all with glacial-geological literature relevant to glaciomarine processes and sediment deposition in a grounding zone and ice shelf environments. Such process literature is key to understanding the sedimentary structures imaged in the seismic data. Without reference to such literature you cannot make the link between the present-day ice shelf channel and the sediments beneath the sea floor. Though there are clearly more modern literature available, a good place to start would be David Drewry's textbook on

Glacial Geological Processes (1986). What are the processes that the seismic observations of the cavity and the subsea sediments give insight into? What might be the glaciological processes that determine sedimentation in ice shelf cavities and at grounding lines?

*General comments 1:*

*Thank you for making this point. We overlooked this and will back this up with literature and a more focused interpretation. We'd like to stress that we make an interpretation of the radar and seismic profiles so at best evidence is circumstantial. However, we do believe the interpretation we provide is the best explanation of what we observe in the seismic and radar profiles. This is also supported by the evaluation of Adam Booth, reviewer #1, one of the seismic experts in glaciology.*

*In our answer we'll use the following terminology:*

*Grounded ice:*

- *Subglacial channel: a feature between the ice and the bed, probably water filled. Needs ice to be visible.*
- *Landform: a geomorphological feature of the bed, would be visible without the ice*

*Ice shelf:*

- *Surface channel: Meandering narrow long channel at the surface of an ice shelf*
- *Basal channel: The sub-ice shelf channel causing the surface channel through hydrostatic adjustment.*

*The seismic survey concentrates at a surface channel caused by a basal channel at the ice shelf of Support Force Glacier. The basal channel is formed upstream by a subglacial channel we see in radar profiles. At the grounded ice we can track the subglacial channel at radar profiles 3, 4 and 5. At profile 3, 7.1 km upstream from the grounding line, the subglacial channel is hardly distinguishable from the bed after which its height increases at profile 4, 4.4 km upstream from the grounding line, to approximately 100 m above the surrounding bed. At profile 5, 1.8 km upstream from the grounding line, the top of the channel increased to approximately 250 m above the surrounding bed.*

*Profile 6 lies at the grounding line: the western part has passed the grounding line, the eastern part has not. The basal channel, an extension at the ice shelf of the subglacial channel, now reached a height of approximately 300 m above the surrounding base of the ice shelf.*

*This is where we'd like to adjust our interpretation: Considering the comment of reviewer #1 that the reflection coefficient of the off-nadir reflections is tricky (we don't think they are diffractions) and we might over interpret the data, we will only use its polarity which indicates the presence of water. The radar profiles show the subglacial channel increases its height from approximately 0 m to 300 m over a length of 7.1 km approaching the grounding line. This would place a landform within 7.1 km upstream of the grounding line which we think is unlikely. Summarizing, if we leave out the value of*

*the reflection coefficient we see no evidence the off-nadir reflections in seismic profile I between radar profiles 5 and 6, are caused by a landform.*

*We interpret these reflections to come from the subglacial channel we see in the radar profiles 3, 4, 5 and 6. The increase in size, when approaching the grounding line, is likely caused by the ocean is interacting with the subglacial channel due to tidal motion thereby increasing its size due to melting of the channel walls as suggested by Drews et al. (2017), Horgan et al. (2013) and modelled by Walker et al. (2013). The radar profiles 3, 4, 5 and 6 show the subglacial channel interacting with the warm ocean. Once passed the grounding line this wide opening of the subglacial channel adjusts to hydrostatic equilibrium and forms the basal and surface channel in which the subglacial drainage water incises.*

*We plan to adjust our interpretation accordingly: At the grounded ice of Support Force Glacier radar profiles 3, 4, 5 and 6 show a subglacial channel connecting a basal at the grounding line. Approaching the grounding line the subglacial channel increases its size to 300 m height at the grounding line, which we attribute to ocean interaction. This setting is similar to the subglacial estuary described by Horgan et al. (2013). Because the subglacial channel connects to the only basal channel at the western side of the ice shelf, and because we have a large subglacial drainage influx modeled at the western side of the ice shelf, we interpret the subglacial channel to be a subglacial drainage channel.*

*The grounded part of profile I consists of a sediment layer judging by its reflectivity becoming more consolidated closer to the grounding line. So the drainage channel probably travels over a layer of subglacial sediments with varying consolidation. The exact nature of the subglacial drainage system we do not know but the radar and seismic profiles do suggest channelized flow close to the grounding line. Possibly we are dealing with a channel that, upstream and outside the survey area, is coupled to a surrounding distributed system as described by Hewitt (2011). Close to the grounding line channelized flow is favorable which corresponds to our observations.*

2. I am not, at present, convinced that the observations of the stratified sediment beneath the ice shelf channel have any bearing on the ice shelf channel and modern- day "sediment transport" itself. The manuscript makes no convincing case that the sediments were deposited by present-day processes. These sediments could be much older than the ice shelf channel and may have absolutely no relationship with modern- day processes at the grounding line or beneath the ice shelf. The authors need to either (1) provide a much stronger justification for the direct link between the sediment and the modern-day glaciology (e.g. by using the literature on glaciomarine sedimentary processes I refer to above and/or better describing and presenting the data they report). It is not enough to simply say on page 21 that "we conclude the landform is hosting the transport of sediments that are deposited in the ocean cavity close to the GL" – what is the evidence?; or (2) reframe the paper so that it is a detailed characterisation of the form and physical properties of (a) the ice shelf channel; (b) the sub-shelf bathymetry; and (c) the sub-bottom sediments of SFG, but doesn't link them directly. For what it is worth I think a manuscript describing '2' would be useful and worthwhile. We know so little about Support Force Glacier at present.

*General comments 2:*

*To summarize our findings:*

*We have a modelled large influx of freshwater on the western side of the shelf.*

*From the airborne radar data of the shelf we know the ice shelf has only one basal channel at the western side. That must be the place where the subglacial drainage channel enters the ocean.*

*There is a noble gas sample downstream of Support Force Glacier suggesting a freshwater influx of terrestrial origin coming from Support Force Glacier.*

*Why do we think the water of the subglacial channel carries sediments?*

*Along profile I (along-flow, 1.5 km east of the basal channel) shows an approximately 200 m thick sedimentary sequence close to the grounding line of different character then the seabed further downstream part of the ocean cavity. The sedimentary sequence is less consolidated and has chaotic reflections and little signal loss with increasing depth. Across profile III, crossing this sedimentary sequence with chaotic reflections, shows this sequence is only present under the sub-shelf channel. Both on the far east and west side of profile III there is hardly any structure in the seabed except right under the channel. This sedimentation most likely has been transported by the subglacial channel.*

*Based on profile I and III we interpret the sedimentation to be point sourced and fan shaped, possibly a grounding line fan (Powell, 1990) or an ice-proximal fan (Batchelor and Dowdeswell, 2015). This explains the chaotic reflections (we referred to as disturbed), with high amplitudes in this sedimentary sequence and this material being softer as the further downstream part of the sea bed.*

*We realize there are concerns here as the fan has formed under an ice shelf of Support Force Glacier without surface melt, a characteristic of fans (Powell and Alley, 1997). But we do have evidence for channelized flow at the grounding line, a noble gas sample suggesting freshwater observation influx of terrestrial origin likely (Huhn et al. 2018) and a significant (190 x 10$^6$ m$^3$ a$^{-1}$) modelled channelized freshwater influx at one place on the west side confirmed by the presence of a single basal channel on the western side. We also have an unusual ocean cavity with a steeply descending seabed and, as argued in our paper, a stable grounding line. These are typical conditions for the formation of a fan at the grounding line (Powell 1990, Powell and Alley 1997, Batchelor and Dowdeswell 2015) . We will emphasize this in the text and update figure 4 with a schematic lay out as in figure 2 where we identify the sedimentary sequence with chaotic reflections.*

*Can we proof all this and can we say how old this sedimentation process is? Not without sea bed samples of the sedimentary with chaotic reflections or an additional seismic across-flow profile passing the subglacial channel. Do we think this interpretation is likely and sound? Yes we do if we look at the glaciological setting; a grounding line environment where a subglacial drainage channel enters the ocean cavity with a descending seabed and seismic profiles show a sedimentary with chaotic reflections right under the basal channel.*

3. The assertion in the abstract and section 4.5 that the "landform is hosting the transport of sediments that are deposited in the ocean cavity close to the GL" (page 21) is very poorly supported by any evidence apart from the spatial coincidence between the landform (described in other parts of the manuscript, e.g. section 4.3, as a "subglacial drainage feature") and the ice shelf channel. What is the process that is being inferred here? In the conclusions it is suggested that the sediment

transport is by subglacial meltwater, but that is not developed from a detailed, carefully constructed and coherent argument in the manuscript.

*General comments 3:*

*In our reaction to general comments 1 we explain our adjusted interpretation: the off-nadir reflections are probably caused by the enlarged opening of the subglacial channel, not the landform. In our reaction to general comments 2 we explain why, based on seismic profile I and III we interpret the sedimentary sequence with chaotic reflections to enter the ocean cavity through the subglacial channel.*

Please don't get me wrong, I think the manuscript is full of great data and interesting observations, but at present it seems to lack a clear focus, and many of the assertations are not fully thought through or developed from a process-oriented perspective. It's all very well to justify the landform identified in the ice-penetrating radar as being comprised of sediment from the seismic reflection coefficient analysis (though it is clear that the seismic data are not perfectly acquired to do this), but the manuscript makes some very large leaps from 'this is a subglacial sedimentary landform (with possibly a bedrock core)' to the stratified sediment offshore is the direct result of focused melting from the ice shelf channel. What is the process by which this happens? It needs to be justified.

*We agree we should make a better case here. We've set out our reasons as to why we think we can connect the sedimentary sequence with chaotic reflections we see in seismic profile I and III to the subglacial channelized flow and why we think this sequence probably resembles a fan.*

An idealised conceptual model of the ice shelf cavity/grounding line/ice shelf channel processes and environments might be useful. See examples in Le Brocq 2013; Drews et al. 2017; and Jeofry et al. 2018 (Suppl Info). A conceptual model like these would really help pull together how the SFG system works and make the manuscript far more accessible to prospective readers. I suspect that developing one might also help the authors think through the processes and allow them to piece together a much more coherent argument and explanation for their observations.

*Conceptual model: Is this like a picture explaining the model? It should be possible but it is quite some work, There are figures showing the formation of fans at grounding line like in Powell (1990). The difference in our case is that there will be a shelf at the grounding line instead of a cliff. But if you feel the paper needs it, we can provide it.*

Please note that there are a few typos and minor grammatical errors throughout the manuscript that will need correcting before publication. I have highlighted some below, but I have not been comprehensive in this.

Specific comments

Title: Support Force Glacier is East Antarctica, not West.

*Title: Not changed as we stick to the definition that the EAIS lies between 45° west and 168° longitudinally.*

Title: Where is the evidence for "sediment transport" in the paper?

*Title is changed according to our adjusted interpretation.*

Addresses: 'Natural Environment Research Council' (i.e. not National Environmental).

*Addresses: Updated*

Abstract (L1): "surface channels" not "flow stripes". Flow stripes are something different, and are not associated with basal ice shelf channels.

*Abstract L1: Agreed, the surface channel at Support Force Glacier starts as a meandering surface channel at the grounding line, is not a flow stripe.*

L5: "beneath" rather than "on"? L6: "part" of what?

*L5: Corrected. Floating part should be ice shelf*
L8: "initiates" rather than "forms"?

*L8: Agreed*

L10: What is the justification for the 200 m thick sequence of sediments being "grounding line deposits" – what is the evidence and argument for this? At present the manuscript doesn't provide this.

*L10:This is an interpretation we give in our reply to the general comments. We will adjust the discussion text accordingly and will explain why we interpret this sequence as a grounding line fan.*

L10: "the landform hosts the subglacial transport of sediments" – why not just "the landform is composed of sediments". That seems to be as far as you can take the interpretation of the seismic data as far as I can make out from the manuscript. There is no evidence for subglacial transport of sediments from the data presented.

*L10: This will be removed as we interpret this feature no longer as a landform.*

L15: "shear margins" are introduced here, but figures 1a&1b suggest that the sub- glacial hydrological pathway does not correspond to the shear margin.

*L15: Indeed the channel is 4 km east of the shear margin. We will remove this.*

L4: surface expressions of basal ice shelf channels are not the same as flow stripes, and sometimes they 'jump' across flow stripes. They are surface features associated with linear depressions in the ice shelf surface.

*Page 2*

*L4: We'll remove the association with flow stripes: "They are often detected with satellite imagery like MODIS..."*

L25-27: some misrepresentation of Jeofry et al. 2018 here – the hard rock landforms at Foundation Ice Stream (FIS) actually determine subglacial hydrological pathways & it is the basal water that forms the ice shelf channels, not the bedrock bumps per se. The marine landforms presented in Jeofry et al. 2018 did not "confirm" anything. That particular figure was simply included to demonstrate how common such hard bed landforms were offshore and presenting them as a plausible analogues for the ridges beneath the FIS grounded ice.

*L25-27: We disagree. Jeofry et al.2018 suggest a combination of a landform incising the base of the ice which when becoming afloat will cause a surface channel and basal channel. Quote: "we propose that the bedforms are dictating the position and form of the U channels." Which is also why they checked the dimensions of landforms that are indeed at completely different locations which we also state in L26, 27. As the landform also organizes the drainage pathway, quote:"the water incises upward into the corrugation peak" also because fresh water will want to move upward will assemble in the by bedrock formed corrugation peak.*

L29: I do not understand "become spots at the ice shelf base when adjusting the hydrostatic equilibrium"

*L29: The surface trough of the shear margin (that has a surface depression) induces a basal channel due to hydrostatic adjustment once it passes the grounding line. Once afloat, the surface trough is shallower while adjusting but then deepens again as a warm water plume thins the base of the ice in the channel.*

L32-33: I do not think that this manuscript provides information on the "type of material and structure of the bed upstream of a basal channel". Yes the apparently offline seismic reflection in figures 2&8 is analysed to suggest that it is composed of unconsolidated sediments, but it certainly doesn't reveal the structure of the subglacial bed.

*L32-33: Indeed, we state that this observation is often missing, nothing more.*

L4: it is interesting to note that the modelled subglacial outflow in figure 1b does not map exactly against the ice shelf channel (i.e. they are offset by ~5 km despite a new high-resolution bed topography of the SFG trunk).

*Page 3*

*L4: This is correct, the modeled drainage pathway is offset by 4 km from the basal channel. This model is coarse, it has a resolution of 1 km and does not take the physical nature of the bed into account that may steer the pathway somewhat differently. Although we did use the topography derived from the airborne radar data, the surrounding is of course still BEDMAP2. So the model is an indication of where one may expect a subglacial drainage system.*

L8: "into the precise"

*L8:Agreed*

L9: "typically penetrate"? There are some examples.

*L9: Agreed.*

L11: I do not understand "..or does the substrate also consist of sediments in which case we can expect recent sedimentation on the seabed?" – even if the bed is hard bedrock you can still have sedimentation on the seabed by (a) subglacial erosion of bedrock; and (b) transport of sediment from upper parts of a glaciers catchment underlain by sediment.

*L11: Thanks for making this point, we will rephrase "or does the substrate also consist of sediments."*

L16: "supported" rather than "backed up"?

*L16: Agreed*

L17: "continue" rather than "proceed"

*L17: Agreed*

L18: "active/current" subglacial drainage?

*L18: Agreed*

L23-24: Support Force actually lies between Academy Glacier and Recovery Glacier. It may also be worth making clear that "northwest" is "grid northwest".

*L23-24: Thanks, corrected*

L24: "constrained" rather than "tugged in"?

*L24: Agreed*

L30: reference Bedmachine as well/instead of Bedmap2?

*L30: Just BEDMAP 2.*

L33: I would not describe the survey as a "grid". Refer to figure 1 at the end of line 33.

*L30: Agreed*

L2: "Ice surface velocities"?

*Page 4*

*L2 : Correct, thanks*

L11: I do not believe that the DLR acronym has been defined earlier in the manuscript.

*L11: We will correct this*

L14: Jeofry et al. ESSD, 2018 might be a better reference than Rippin et al.? see
https://essd.copernicus.org/articles/10/711/2018/

*L14 Jeofry et al: Good suggestion, thanks*

L14: think you mean 312.5 Hz here? See section 2 of
https://agupubs.onlinelibrary.wiley.com/doi/full/10.1029/2018GL077504

*L14: Correct*

L16: "laser surface terrain"?

*L16: Correct*

L27: Paden et al. reference is 2010 in reference list? Please also be specific about which years of OIB data were used (e.g. up to Dec 2018 Antarctic surveys)?

*L27: Thanks for bringing it to our attention. The reference should be:*
*Paden, J., Li, J., Leuschen, C., Rodriguez-Morales, F., and Hale., R.: 2010, updated 2018.*
*IceBridge MCoRDS L2 Ice Thickness, Version 1. Antarctica., Boulder, Colorado USA. NASA*
*National Snow and Ice Data Center Distributed Active Archive*
*Center.,https://doi.org/10.5067/GDQ0CUCVTE2Q, 2019.*

L30: how extensive was the model domain for the hydrological routing presented in 1b? Was it just the domain shown in 1b, or was it more extensive (e.g. entire SFG catchment)?

*L30: The model domain for the routing was the entire hydrological catchment of SFG. I attached a small figure, which we could put into the answer or appendix.*

[Figure]

Figure 1: Please annotate start and end of seismic lines in figure 1 and corresponding seismic profile figure.

*Page 5*

*Figure 1:We provided arrows at each line (Fig 1b).*

L1: as stated previously, the modelled subglacial meltwater outflux location does not correspond that well with the surface channel, despite the new high resolution ice thick- ness/bed data.

*L1: As explained (Page 3, L4) we use a model with its shortcomings. The main result is we can expect water drainage on the western side of the shelf. Keeping in mind that the resolution of the model is 1 km, and does not take the physical properties of the bed into account, we find 4 km acceptable.*

Table 1: Why is it important to have the column "position from GL"? From which part of the profile is this measured? Suggest deleting it.

*Page 6*

*Table1: We will remove this*

Pages 7-9

Given the expertise of reviewer 1 I have not assessed the description of the seismic methods in detail.

L16: After "ice shelf thickness." it might be a good idea to refer to figure 1 to provide an example of the artefacts.

*Page 10*

*L16: Reviewer #1 pointed out that the phrase artefact is not accurate. We'd like to change the title to "seabed depth conversion" as he suggests.*

L18: "structure of the seafloor" or "morphology of the seafloor"?

*L18: The morphology is a better phrase*

L20: Figure 2 should be split into 2a and 2b, rather than TOP and BOTTOM. Therefore, sentence should begin "Figure 2a shows. . .."

*L20 Figure 2: Agreed*

L20 (and throughout manuscript): Delete references to along-profile and across-profile, just label the profiles I-V (e.g. "profile I" rather than "along-profile I"). L21: Start "Figure 2b. . ."

*L20: Agreed*

L24 (and throughout manuscript): "reflectivity zones" rather than "intervals". These zone should also be clearly marked on figure 2.

*L24: They are marked with double headed arrows on top of the figure: interval 1, 2, 3 and 4.*

L28: call figure 1 after "interferometry"

*L28: Agreed*

L30: "uncoupled" – uncoupled from what? Do you mean "floating"? Page 11

*L30: Yes we mean floating*

Figure 2: label 'a' and 'b'. Need to annotate the start and end of the seismic line (see comments on figure 1).

*Page 11*

*Figure 2: labeling agreed. The shot numbering gives the shooting directions, but we can add this if you feel this is not clear.*

Note that where "bed" is annotated in 2a it is not actually the 'bed'. It is the subsea sediments.

*Note: Is seabed acceptable? If we talk about sediments we are interpreting.*

What is the weak reflection between ~SP46 and ~SP160 (between the labels "ICE" and "BED")?

*Weak reflections: The are probably side reflections of the ice-sea contact, the polarity is reversed just as the identified seabed contact which is why we think they resemble ice-seawater contacts. The shelf base here has a lot of topography*

Please define the different reflectivity zones in figure 2 (see comments above). Provide zoomed-in views of all the detailed features described in section 3.2.

*Reflectivity zones: As mentioned we have defined the intervals and they are based on structure on the bed, ice-sea contact and/or seabed. We find reflectivity does not cover the separation of the intervals properly. Please let me know if you find this ok. We can provide zooms of key features.*

L1: "….elongated feature above the flat bed…."?

*L1: Based on shot spacing and two-way travel time the dimensions are 1200 m long and the feature appears to be 50 m higher than the surrounding bed if it was nadir, hence we called it elongated although of course we do not know the across-flow dimension.*

L3-4: I suggest that this reflection not be described as a "subglacial drainage feature" at this point in the manuscript. Later in the paper, the reflection is interpreted as a "landform" anyway, so it is very confusing. Simply describe it a reflection at this stage, and then only once section 4.3. has been worked through should it be described as a "landform". It would be useful to have the zoom in of the reflection in figure 2 rather than figure 8.

*L3-4: We propose to call this "subglacial feature". We will provide a zoom of this subglacial feature in our response to reviewer#1 as we think we are dealing with reflections. As mentioned this part of our interpretation we want to change: We interpret this subglacial feature still as off-nadir reflections but no longer as a landform but as the top of the subglacial channel that in this area so close to the grounding line likely interacts with the ocean. This interaction with the ocean probably caused a rapid increase in height of the subglacial channel.*

*As reviewer #1 pointed out, a quantitative analysis its reflectivity is tricky as we have a complex subglacial structure off-nadir. To avoid over-interpretation we will not use the calculated reflection coefficient but its polarity.*

L6 (and throughout manuscript): "anticlines" – this is a specific geological term not normally used in the description of morphology. I recommend "concave cavity" instead.

*L6: Agreed, concave cavity is a better term.*

L8: "ocean cavity thickens" (rather than deepens)?

*L8: Agreed*

L10: what is the observation that constrains the sediment thickness to 200 m? I don't see a clear sediment-bedrock reflection at 200 m in Figure 2, so do the authors instead mean "of at least 200 m"?

*L10 200m: It is as you observe, there is not a clear last sediment-bedrock reflection but the chaotic reflections fade out with increasing depth. Hence approximately 200 m.*

L10 (and throughout manuscript): what is meant by "transparent"? How can a material be transparent yet stratified and disturbed?

*L10 transparent: What we mean by transparent is that the seismic signal penetrates deep in the formation with little loss of amplitude.*

*We will use the phrase (sedimentary) sequence with chaotic reflections and little signal loss with increasing depth or shortened: sequence with chaotic reflections.*

*as mentioned in our reaction to general comments 2.*

L15: Call figure 1 after "at the same distance"?

*Page 12*

*L15 Agreed*

L20: call figure 3 after "have been marked"?

*L20: As this a complicated profile with two ice-sea contacts and two sea-seabed contacts, we'd like to take this figure out of the paper as reviewer 1 suggests. We hardly use it in our interpretation and the figure is complicated to explain.*

L21: After "the seabed" add "so is not presented here"?

*L21: Yes the seabed is present here as we have the seafloor returns twice (two different ray paths: path 1 is along crest of the channel, path 2 is along base of the ice next to the channel and they likely have the same seabed depth. Converting the time migrated section to depth is not really possible as we must choose one of these two ray paths to convert to depth but then we automatically misplace the reflections of the other ray path.*

L22: "sequence of stratified sediment" rather than "stratification sequence"? If the authors are trying to avoid interpretation here, then it shouldn't even be "stratification sequence", it should be a geophysical description like "a series of horizontal reflections".

*L22: We will remove Figure 3 as profile II is difficult to interpret and probably causes misunderstandings. As reviewer #1 pointed out, profile II hardly contributes to the interpretation.*

L23: "of stratification below the basal channel" rather than ".. of a stratification sequence in the basal channel". The seabed and sub-bottom sediments are not in the basal channel.

*L23: Will be removed*

L24: call figure 3a at end of sentence.

*L24: Will be removed*

L31: "terraced" not "terrace-shaped"

*L31: Thanks*

L32 (and throughout manuscript): instead of "as indicated in Figure 4" just have "(Figure 4)". There are a lot of wasted words throughout the manuscript when fig- ures are being called. Authors should make their statement/describe the data etc. and then simply cite the relevant figure at the end of the sentence. See Box 1 of https://aslopubs.onlinelibrary.wiley.com/doi/full/10.1002/lol2.10165 for a better explanation of what I mean. There is also no need to repeat figure caption information in the text (e.g. page 12, line 26 opening line of section 3.4).

*L32: Thanks for pointing this out*

Figure 3: what are the reflections above the seabed? It is not clear in the figure what all the complex reflections are. More annotation is required.

*Page 13*

*Figure 3: There are two seabed reflections present due to different travel paths. See our reply at L21. Anyway we will remove Figure 3*

Figure 3: Again, authors should not refer to sub ice shelf sediments as "bed". Bed should only be used when referring to grounded ice.

*Figure 3 bed: We used the term seabed which we do find in literature.*

L3-7: This paragraph should refer to figure 4.

*L3-7 Agreed*

L6-7: I do not think that I understand the sentence "The across-profiles. . .. . ..across- profile III". I have a suspicion, however, that this sentence might actually be quite key to the authors' suggestion that there is a relationship between the ice shelf channels and the sediment beneath it. If I understand it correctly, here they are suggesting that there is a thicker stratified sequence in the

parts of the sub-sea sediments directly beneath the ice shelf channel. Is that correct? At the very least, the authors need to annotate the thicker sequence below the ice shelf channels to assist the reader understand exactly what is being described here.

*L6-7: Yes that is what we claim, right under the basal channel, profile III shows thicker stratification (roughly from SP 3 to SP 24) under the basal channel then outside the basal channel. We plan to add schematic images (recommendation of reviewer 1) of profiles 3,4 and 5 as in Figure 2 marking the stratification areas.*

Figure 4: This figure appears to be critical to the argument that the sediments beneath the ice shelf channel were deposited by the ice shelf channel (i.e. there is a spatial coincidence between the channel and a thick sequence of sediments subsea). What is the thickness of the sediment package beneath the ice shelf channel in figure 4a (note that figure subplots need to be labelled throughout the manuscript)? Is it, as implied by the y-axis, 400 m? If so, that is a phenomenal amount of sediment to be deposited 40 km from the grounding line solely from melting beneath an ice shelf channel. The authors should calculate the sedimentation rate for this package of sediment. Is it possible for their sedimentation rate to be valid (e.g. assuming a certain proportion of sediment in the ice, and known melt rates for ice shelf channels as calculated from ApRES). Figure 4 needs to be better described in the text and the key features (e.g. sediment packages) need to be better annotated. I assume that the ages of transit time from grounding line are based on current ice velocity (i.e. figure 1a)?

*Page 14*

*Figure 4: Indeed I see the confusion. Profile V (as profile IV) has multiples causing apparent stratification. This is clearer to spot in the time migrated profiles. So no, there is not 400 m of stratification at profile V (Figure 4a), I come to no more than 100m. We will provide a schematic picture with our interpretation. The focus of the paper lies on the sedimentary sequence with chaotic reflections that profile III crosses.*

Figure 5: why not just make this a 3D figure and show all 4 profiles (figure 2b of https://agupubs.onlinelibrary.wiley.com/doi/full/10.1029/2010GL042884 is an example of what I mean)?

*Page 15*

*Figure 5: Thanks for the suggestion. What we like to provide is use both radar and seismic profile to show the development of the subglacial channel (grounded) and how this continues as a basal channel under the ice shelf.*

*The present figure actually consists of 3 profiles, profile 4 is used twice. Reason why we displayed them like this is to get a good handle on where melt/widening of the basal channel takes place.*

L5-12: There is lots of text in this paragraph that is direct repetition of the figure caption of figure 6.

*L5-12:We will clean this up*

L5: "We selected ten across ice radar profiles. . .."?

*L5: Agreed*

L6: what is the evidence for the subglacial landform "shaping the channel upstream of the GL"? How does it do this, and what is the process?

*L6: We withdraw this interpretation*

L11-12: This sentence needs some expansion to make 100% clear the spatial coincidence between the landform and the ice shelf channel. Reference to figure 1 would help too. I don't understand the phrase ". . . after which the landform become indistinguishable from the bed"? Surely if it is a basal landform it is the bed? It would also be useful to get a better idea and description of the wider bed topography around the landform (e.g. entire bed of SFG) to understand the context. Figure 1b is of little use in this regard its colour scheme is very uninformative.

*L11-12: We will adjust our interpretation as we state in our reaction to general comments 1 and remove the concept landform. Looking at the radar profiles 3, 4, 5 and 6 we see that the subglacial channel we see at the grounded ice, increases in size as it approaches the grounding line. So there is no landform at the grounded ice, just a subglacial channel that increases its size due to interaction with the ocean.*

Line 15: again, lots of text that should be in the figure caption, or already is. Page 16

*L 15: Agreed*

Figure 6: Indicate very clearly which radargrams are over grounded ice and which ones are over floating ice.

*Figure 6: Agreed that should be made clearer. Profile 6 lies at the grounding line.*

Figure 6: are the authors absolutely sure that radar profile 5 is fully grounded all the time? Could there be tidally-induced grounding line migration?

*Page 17*

*Figure 6 profile 5: We have no indication profile 5 is susceptible to grounding line migration. Profile 5 crosses seismic profile I at SP 5 where we have a positive basal reflection indicating*

*consolidated material. To us that means the ice is grounded here. If ocean water would have reached this far it would have influenced the reflectivity.*

*We do have an indication the MOA grounding line, crossing seismic profile I at SP 51, is not correct. Seismic profile I clearly shows ocean water being present upstream of the MOA grounding line down to SP 26.*

*Are we sure profile 5 is fully grounded all the time? No but it is very likely.*

Section 4.1: It is a little difficult to link section 4.1 with figure 1, as the text in section 4.1 refers constantly to shot points, but these are not apparent on figure 1.

*Section 4.1: We will be clearer here. We wish to refer to figure 2, profile I here, and will add this in the text. The interferometric grounding line crosses profile I at SP 23 but this can't be chosen that precise, . The polarity switch at profile I lies at SP 26, so 150 m downstream of SP 23. This deviation may be caused by the unprecise choice of the grounding line here.*

Line 14: "topographically constrained flow"?

*L14: Correct*

Figure 7: it took me quite a while to figure out exactly what this figure was. I suggest that it simplified by removing the bed profile picks. The key point of this figure is to conceptualise the idea of the offline reflection. Is the red semi-circle a "possible drainage feature" or it is a "landform"?

*Page 18*

*Figure 7: We think the concept of Figure 7 is still not clear.*

*The figure should show that off-nadir reflections of the landform (represented by the radar profiles 5 and 6 and we now interpret as the subglacial channel) arrive at the same time as if there had been a 50 m high channel at nadir (represented by the red semi-circle). As reviewer #1 points out, the weakness of the reflections shown in figure 8 (a zoom of profile I, figure 2) already suggest these reflections (or diffractions as reviewer 1 points out) are off-nadir.*

Section 4.3: I recommend not describing the feature being evaluated as either a "subglacial drainage feature" or a "landform" until the authors actually determine which of the two hypotheses are their preferred one. Whilst the unusual (offline?) reflection they describe is referred to throughout as a "subglacial drainage feature" the author then state on page 19 lines 16-17 that they "prefer interpretation 1" which is that the reflection is from the landform. This is a bit of a mess, and suggests that the authors have changed their preference during the writing of the manuscript but not updated all parts of the manuscript. A "landform" is not a "subglacial drainage feature".

*Section 4.3: Indeed, we adjusted our interpretation as described in our reaction to general comments 1 and will adjust the text accordingly.*

Section 4.3: change heading to "Does the seismic data record a subglacial drainage feature or a subglacial landform?" or something along those lines. Section 4.3 evaluates these two hypotheses on the basis of the seismic data and geophysical theory. The section heading should reflect that in some way.

*Section 4.3: We will restructure this according to our interpretation: The reflections are off-nadir and represent the subglacial channel. The channel opening is enlarged here due to interaction with the ocean. This interaction between ocean and a subglacial channel is described by Horgan et al. (2013)*

Line 13: provide some additional detail about what is meant by a "separate drainage feature on a hard bed" – In essence this is a Röthlisberger (R-) channel incised into the overlying ice and should be described as such here.

*L13: Indeed if the reflections are at nadir it would seem like an R-channel and that is represented by the red semi-circle. That this is most likely not the case is because there is only one basal channel visible at the western side of the ice shelf and we argue that this is where the subglacial channel enters the ocean cavity which is on the western side so off-nadir of profile I. Had the reflections been nadir, the R-channel would have entered the ice shelf elsewhere but we see no evidence of another basal channel in the radar data. That is our main argument as to why we think the reflections are off-nadir and are caused by the subglacial channel.*

L17-20: This is an important admission here, and one that is entirely inconsistent with the title of the manuscript "Subglacial sediment transport upstream of a basal channel. . ...". So far, the data don't even unequivocally demonstrate the presence of subglacial sediments.

*Page 19*

*L17-20: Correct, it is an interpretation.*

L14: "floating ice" rather than "uncoupled ice"?

*Page 20*

*L14: Correct*

L16 (and throughout manuscript): what do the authors mean by "disturbed"? This needs defined and highlighted/annotated in a figure. Do the authors mean "deformed" sediments or stratigraphy?

*L16: We propose chaotic reflections and little signal loss with increasing depth, as mentioned in our reaction to general comments 2.*

L17-19: apart from the reflection coefficient values and the "disturbed and stratified" stratigraphy, what other lines of evidence for these materials being grounding line de- posits do the authors have? This is where reference to glacial geological literature is essential.

*L17-19: We are presenting an interpretation. Seismic profile I and seismic profile III most likely show the presence of a grounding line fan.*

L31: ". . .are positioned on the western side of SFG near its shear margin."?

*L31: Agreed*

L32: I disagree that you can track the landform at least 7.7 km. It is not apparent in profile 3 (Figure 6), so it can therefore be tracked for a maximum of 5.2 km (i.e. up to profile 4).

*L32: If you follow the same flow line along profiles 3, 4, 5 and 6 (marked on the long profiles with an arrow and radar trace number it is quite obvious. We also have 38km long profiles that make a clearer case for this observation which we can provide possibly in a supplement.*

L33: "some degree of consolidation" – so is it unconsolidated sediments, or not?

*L33: As we pointed out in the our reaction to general comments 1 we will withdraw the quantitative analysis of the reflectivity. We will just use the polarity of the off-nadir reflections.*

L1-23: I am totally unconvinced at present by the argument the authors make linking the sub cavity sediment stratigraphy with the ice shelf channels. I see no evidence at all that (a) sediment is being discharged at the grounding line from a subglacial hydrological channel; or (b) that sediment is being deposited directly from basal melt within the ice shelf channel. There seems to be a huge leap of faith being made in this part of the discussion, particularly in L4-5 i.e. "Taking the evidence together we conclude the land- form is hosting the transport of sediments that are deposited in the ocean cavity close to the GL" – I see absolutely no evidence presented in this manuscript supporting the transport of sediments. If the authors wish to pursue this angle in a revised manuscript then they will also need to explain the transport mechanism. Is it subglacial deformation advecting sediment to the grounding line? Is it sediment transport by subglacial meltwater? Or, is it englacial sediment transport and then melt out?

*Page 21*

*L1-23: There is clear evidence of subglacial drainage at the western side namely the basal channel itself which matches a modelled subglacial drainage pathway with a large water flux. The radar profiles 3, 4, 5 and 6 indicate the presence of a subglacial channel matching the location at the grounding line of the basal channel. The increase in height of the subglacial channel seen on profiles 4,*

*5 and 6, close to the grounding line can very well be explained by interaction with the ocean. This is what one would expect of channelized flow close the grounding line and has been suggested by Horgan et al. (2013) and Drews et al. (2017) and modelled by Walker et al. (2013) and Hewitt (2011). What we can't proof is that this channel is carrying sediments but it is likely that at the end of an ice stream the subglacial channelized drainage system carries sediments. We do have the observation in seismic profiles I and III, of a sedimentary sequence with chaotic reflections close to the grounding line (profile I) and the presence of this package only under the basal channel (profile III), exactly where one would expect sedimentation to take place if the subglacial channel would be carrying sediments.*

L12-14: Here, the authors state "What we do see is more stratification in all across- profiles below the sub-shelf channel than outside of it and that this stratification extends to the eastern side of across-profile III." If this is the case then this is potentially important, but at present (except perhaps a hint on page 13) this observation is not effectively presented in the current version of the manuscript. This needs strengthened considerably if the authors are to underpin their argument robustly. I remain unconvinced though that their survey layout is extensive enough to permit this statement. I would also like to see an assessment of the implications of thinner ice (and therefore < englacial signal attenuation) over the ice shelf channels – could this lead to higher amplitudes reflections from subsea interfaces beneath the channels compared to the sediments beneath the thicker ice beyond the channels? I also think that the authors need to carefully consider the entire sediment package (i.e. the stratigraphic relationships between the sediment beneath the ice shelf channels and those beyond the channels).

*L12-14: Profile III shows thick sedimentation only under the basal channel consisting of several levels and extending eastward. We agree we should emphasize this observation and it's interpretation more. This is what links the sedimentation to a grounding line fan where the subglacial channel enters the ocean cavity and forms the basal channel by adjusting to the hydrostatic equilibrium.*

*Profile IV and V have also show sedimentation but are tricky as multiples occur between stronger reflections. See my reaction to your comments at Figure 4. These profiles also cross different formations that are beyond the focus of the paper.*

*When calculating a reflection coefficient, the attenuation in ice and seawater over the entire travel path are taken into account as is pointed out in chapter 2.6, equation 2. As such reflectivity is compensated for the attenuation.*

L29: My understanding of reflection coefficient analysis is that it characterises the physical properties of the upper few metres below the interface. As such it is a stretch to state "the eastern side of the landform consists of sediments. . .". Perhaps make this statement more specific (e.g. "Reflection coefficient analysis indicates that the upper few metres of the landform is unconsolidated sediment. . .")?

*L29: The reflection coefficient characterizes the interface between two media but if there is a layered sequence the reflection coefficient can be influenced by interference. We will just stick to the polarity of the off-nadir reflections.*

L31: OK, so here, finally in the conclusions section, the authors are specific about the actual process they believe is at play, i.e. "The landform hosts a channelized subglacial drainage which transports sediment downstream". I therefore ask the following questions (1) what is the evidence for subglacial drainage? (2) how does a landform "host" channelized subglacial drainage? (3) where is the evidence for sediment transport?

*L31: Evidence for subglacial drainage I pointed out answering your comments at page 21, L1-23*

*The subglacial feature is most likely the subglacial channel interacting with the ocean as pointed out in the our reaction to general comments 1.*

*Profiles I and III are evidence of a grounding line fan under the basal channel.*

L32: How do the authors know that the 200 m thick sediment package has any association with the current processes at the grounding line? These sediments could be ancient and have nothing to do with modern-day grounding line processes. The spatial relationship could merely be coincidence.

*L32: As mentioned quite extensively in our reaction to general comments "we do have evidence for channelized flow at the grounding line, a noble gas sample suggesting freshwater observation influx of terrestrial origin likely (Huhn et al. 2018) and a significant (190 x 10$^6$ m$^3$ a$^{-1}$) modelled channelized freshwater influx at one place on the west side confirmed by the presence of a single basal channel on the western side.*

*Seismic profile I and III suggest the sedimentary sequence with chaotic reflections is point sourced and fan shaped, possibly it is an ice-proximal fan (Batchelor and Dowdeswell, 2015). This explains the chaotic reflections and this material being softer as the further downstream part of the sea bed.*

*We also have an unusual ocean cavity with a steeply descending seabed and, as argued in our paper, a stable grounding line. These are typical conditions for the formation of a fan at the grounding line (Powell 1990, Powell and Alley 1997, Batchelor and Dowdeswell 2015) ."*

*Lastly we do not provide hard evidence but an interpretation.*

L18: Since reviewer 1 has sung the praises of Bradley Morrell, I will sing the praises of Dave Routledge - A brilliant field guide - he's also great!

*Page 22*

*L18: Indeed*

Final comment: I appreciate that the majority of comments above will be viewed by the authors as perhaps overly negative. However, I do want to emphasise to the authors that I have provided the comments above because I feel that the acquired data are excellent and potentially very important.

I would certainly like to see these data and results being published in some way, but I do believe that a stronger more care- fully thought-through and coherent argument needs to be developed to place the assertations and findings put forward on a more secure foundation. I do hope that the comments provided above will assist the authors to achieve this.

Dr Neil Ross Newcastle University 3rd July 2020

*Final comment: Your comments are highly appreciated. They force us to built up our case better which improves the manuscript. So thank you.*

*Coen Hofstede, August 15, 2020*

**Mayor changes made in the resubmission:**

Title:
Changed, the focus now lies on a grounding line fan below the basal channel of Support Force Glacier

Abstract:
Adjusted accordingly and includes our adjusted interpretation that the subglacial feature represents the top of a subglacial channel

Results:
Figure 3 showing profile II has been removed and is not discussed in the paper.
Figure 5 has been added showing a schematic diagram of the subglacial channel based on the basal ice reflection of the migrated radar and seismic profiles.

Discussion:
Changed the sequence of the items in the discussion. We discuss:
- the grounding line, the ocean cavity,
- the seismic facies of the seabed,
- a shortened discussion of the subglacial feature which we now interpret as the top of the subglacial channel connecting to the basal channel. We removed Figure 8,
- the subglacial hydrology and our arguments why the 200 m sedimentary sequence with chaotic reflections are most likely a grounding line fan

Conclusions have been changed accordingly

[revised manuscript text omitted]

---

## Referee Report (RR1)

[referee-annotated manuscript omitted]

---

## Referee Report (RR2)

**Review of revised version of Hofstede et al., Evidence for a grounding line fan at the onset of a basal channel under the ice shelf of Support Force Glacier (West Antarctica) revealed by reflection seismics (tc-2020-54)**

**General Comments**

The authors are to be fully commended for doing a very effective job of addressing the comments and suggestions from the initial round of reviews. From my perspective, I am pleased to see that the authors have made an attempt to engage with relevant glacial geological literature that addresses many of the comments on the first submission.

There are a few issues that I would like to either re-emphasise, or issues that have arisen because of the changes to the manuscript. This manuscript's significance is that it is the first time that we have seismic observations of the sediments in the vicinity of an active ice shelf channel (making it worthy of publication). I don't dispute the potential for the subsea reflections reported in the manuscript to advance understanding of ice shelf channel processes of deposition, but I remain of the opinion that the authors are currently making slightly unfounded leaps from their (important) seismic observations of the subsea sedimentary structures, to some of their interpretations. As such, I think that the authors should consider carefully how far they wish to push some of their observations, which are prominent in the title of the paper (i.e. "*Evidence for a grounding line fan at the onset of a basal channel….*"). Most of what I critique below relates to the 'interpretation' of the chaotic reflections in section 4.5 of the manuscript, that despite engaging with relevant literature, doesn't do a good job of justifying why 'chaotic reflections' equate to a grounding line fan. An interpretation must be based on good logic and a strong argument, and at present this isn't the case here where there is a leap from 'chaotic reflections' to 'grounding line fan'.

[*Note that after having completed my review it became apparent to me that figure 2 includes what I would argue is some critical evidence, not currently described in the paper, that would very much strengthen the argument for a grounding-line fan. The authors may therefore want to consider some tweaking to figure 2 to present these data more effectively, and the inclusion of some additional text in the results and interpretation. This would bring stronger evidence for their interpretation of the near-grounding line deposits being associated with a grounding-line fan to the fore. Although it is a small part of the paper, since it is specifically mentioned in the title it is important to strengthen this interpretation.*]

**Specific Comments**

1. I remain totally unconvinced under any definition (i.e. glaciological, geological, geographical) that Support Force Glacier can be classified as West Antarctica. It initiates near South Pole, drains through the East Antarctica Ice Sheet (cf. page 3, line 27 of the manuscript), and becomes afloat east of the Dufek Massif of the Pensacola Mountains. Yes, it flows into the Filchner Ice Shelf, but I would not ascribe Recovery Glacier to West Antarctica on that basis. How about a compromise, and just refer to it as in 'Antarctica' in the title?

2. The updated version of the manuscript (and the new title) refers to 'evidence for a grounding line fan'. I cannot see geomorphological (seabed morphology) evidence for the presence of a grounding line fan in the manuscript (on page 10, line 15 the authors describe the seafloor around the basal channel as "fairly flat"), so the evidence for a "fan" is based solely on the seismic-derived subsea reflections ('*post review' update: see* \*). In the case of the latter, the authors need to either strengthen that argument substantively to make the

case for a 'fan', or to reduce the leap in interpretation a little. What is the geological smoking gun in the subsea stratigraphy that justifies this interpretation? Reference to geological literature on seismic stratigraphy may help in this regard. A grounding line fan is a geomorphological landform that may be identified in the geological (geophysical) record from particular stratigraphic structures and relationships between geological (geophysical) units. At present, I cannot see how the seismic data presented in the manuscript justifies such a strong argument that the data provides "*Evidence for a grounding line fan at the onset of a basal channel under the ice shelf of Support Force Glacier*…", the new title of the manuscript. Yes, it's fine to have 'interpretations' but such interpretations must be grounded in either specific observations and/or supporting evidence/analogies from relevant literature. At present, I don't see that a "grounding line fan" interpretation is securely built from the current observations and the literature.

3. The abstract states "*…..below the basal channel, the seismic profiles show an 8 km long, 3.5 km wide and 200 m thick sediment sequence with chaotic reflections we interpret as a grounding line fan deposited by a subglacial drainage channel directly upstream of the basal channel*." (underlining by reviewer). The authors should be encouraged to consider the implications of this interpretation. Firstly, that is a huge volume of sediment (~5.6 km$^3$) deposited by a single subglacial drainage channel. If true, it would indicate a high discharge channel persisting in/around the same location for a significant period of time, and long-term grounding line stability (or some significant flood events e.g. https://doi.org/10.1017/aog.2019.30 - ~60 m of sediment emplaced in a 4 week window). In addition, if that amount of sediment were deposited in that location, why is there not a geomorphic expression of it? Surely the seabed should have a topographic high associated with the unit of sediment deposited by the channel? If it does not, then what has caused the deposition of sediment adjacent to it? If the argument is that the subglacial water is the cause of high sedimentation rates at the position of the thick sedimentary unit with the chaotic reflections, then what is causing high sedimentation rates elsewhere along the grounding line where there are not meltwater portals/an ice shelf channel? The manuscript's abstract suggests "only little basal melting" which seems incompatible with high sedimentation rates elsewhere along the grounding line.

4. The chaotic reflections are critical to the interpretation of the 200 m thick sediment sequence as being a grounding line fan. What is the justification for this interpretation? What is it specifically about the reflections (e.g. thickness, scale, morphology etc.) that provides diagnostic evidence of deposition by meltwater? The authors should be encouraged to justify the description of these reflections a 'chaotic'. Based on figure 2d, to me it looks like the reflections, whilst sloping, curved, disturbed etc. actually have some lateral continuity and stratification, so their description as 'chaotic' may be a misnomer. These reflections are key to the interpretation of the sedimentary unit as a grounding line fan (see point 2 above), but the building blocks from those observations to the interpretation stage are currently weak – better description of the form and stratigraphic relationships of the reflections is needed (in sections 3.2 and 3.3), and then the interpretation of them as representing a grounding-line fan needs to be built on this description supported by reference to the relevant (glacial) geological literature (which the authors do to some extent in section 4.5). To evidence a fan, I would expect seaward dipping reflections in the along ice flow directions, and across flow reflections that showed evidence for channel migration across the fan ('*post review' update: see \*\**).

At present the paper doesn't describe the chaotic reflection zone in detail (e.g. there is little description of them in the results section (3.2/3.3) - either perpendicular or parallel to ice flow - and they are not effectively annotated on figure 3c), and there is a leap in the logic, from (a) chaotic reflections; to (b) grounding line fan; without the necessary steps in-between. What is it about these 'chaotic' reflections that makes them diagnostic of deposition by subglacial meltwater and therefore the argument for them representing a grounding line fan? At present, the manuscript doesn't make that case. The interpretation that the sediments provide "evidence for a grounding line fan" (as stated in the title) cannot be based solely on spatial coincidence and the fact that the deposits are unconsolidated; there may be alternative explanations for those "chaotic reflections" being located where they are (e.g. subglacial deposition and/or deformation prior to grounding line retreat), and it remains entirely possible that these subsea sediments have nothing to do with the processes occurring at the present-day (/Holocene) grounding line.

5. On page 10, line 15 – do the authors mean "subglacial structure"? Would "subglacial stratigraphy beneath the seabed" not be more appropriate?

6. Line 9-10 of abstract. You cannot have bedrock that has been deposited under "different glaciological circumstances" (unless it is from the Ordovician or Carboniferous periods etc.). As such, a re-phrasing of this sentence is needed.

7. Page 3, lines 13-15 – These key questions don't necessarily align with the title and abstract of the paper. Consider adapting?

8. Figure 3c – can the authors annotate the chaotic reflections on seismic profile 3 (as well as on the schematic), as they have for figure 2? The spatial relationship between the ice shelf channel and the chaotic reflections in this figure is critical to the finding of a "grounding line fan" so it needs to be made obvious to the reader. Personally, I found it impossible to see how the authors had extracted the zone of 'chaotic reflections' drawn in the adjacent schematic from figure 3c (seismic profile III). Based on what is presented, the chaotic reflections seem to extend far beyond the lateral limits of the ice shelf channel, which contradicts the statement on page 20, line 23 that they are "only present under the basal channel".

9. Figure 3 – there are very significant differences between 3c and the other two seismic profiles in this figure. The ice-water interface seems to be a more complex reflection, and there is a poorly defined seafloor reflection (in comparison to the strong ice-water reflection and obvious seafloors of 3a & 3b). Could attenuation of signal be part of the explanation for the weaker, more chaotic, subsea reflections of 3c? The data presented in 3c also looks a little noisier?

**'Post-review' Comments**

* actually, looking at figure 2 (2a & 2d) in detail, I think the authors could make the argument that there is geomorphological evidence for there being a grounding line fan (*but they don't currently do this in the manuscript*). Based on figure 2a, the seafloor is clearly dipping seawards within interval II between shots 46 and 112. The authors should make more of this, as it would strengthen the underpinning of their interpretation for a fan being present at the grounding line. Can you include

some back of the envelope calculations of slope gradient etc. of the seafloor and compare to the surface gradient of known submarine fans?

**Also looking at figure 2a, I also wonder if the authors are throwing away some critical observations that would further strengthen their argument (i.e. the dipping reflections between shots 38-53. Are these not seaward dipping reflections in the along ice flow direction that would be diagnostic of fan-like deposition? Or am I making too much of the data? It looks to me like (a) there is a dipping seafloor here; and (b) dipping reflections seawards of the grounding line (located at SP26). This is the part of 2a I am referring to (possible dipping reflections are annotated):

[Figure]

**Dr Neil Ross**
**Newcastle University**
**18th December 2020**

---

## Author Response (AR2)

Bremerhaven, January 25, 2021

Dear reviewers,

Thank you for your comments which helped improving the paper. The minor revisions did not turn out to be so minor after all but I think it was worth it. The main changes are:

- I inserted a new figure, figure 4: (a) showing the seabed of profile II (that was previously taken out but has important clues) under the basal channel and (b) a top view showing the extend of the sedimentary sequence with chaotic to weakly stratified reflections.

- Figure 2d is slightly adjusted such that the seaward dipping reflectors are shown.

- Figure 3d is a zoom of the seabed of profile III.

Essentially our evidence for a grounding line has two main arguments:
- The sedimentary sequence having a chaotic to weakly stratified reflections, probably caused by gravitational flow along the steeply descending seabed. This sequence is very different than major part of the seabed.
- The extent of this sedimentary sequence. Towards the grounding line the outer boundaries converge towards the subglacial channel making the sequence point sourced and fan shaped.

**Reviewer 1:**
Thanks again for your concise comments, they have been entered.

Two remarks:
- Regarding the AVA data from the shot records. Amplitude loss due to geometrical spreading is accounted for by equation 3.

- Depth migration has been tried as well but did not give satisfactory results. The basal topography of the ice shelf (terraces) was affected by the larger concave cavities. Best results were depth conversion of time migrated sections.

**Reviewer 2:**
Our main evidence for the presence of a grounding line fan deposited by the subglacial drainage channel has several arguments:
- The sedimentary sequence with chaotic reflections. This is to be expected as we interpret this sequence as an outwash from the subglacial drainage channel deposited by gravitational flow. And that is a logic consequence considering the steeply descending seabed close to the grounding line seen on profile I. We can now add the seaward dipping reflectors of this sequence.
- The extent of this sedimentary sequence derived from the profiles.

Profile III is problematic to describe the seismic facies in detail. The seabed topography and morphology are significantly influenced by the topography of the basal channel. What is possible is to estimate the extend of the chaotic sequence because of its larger amplitudes and because profile III crosses profile I (the sequence top and base can be clearly defined) which are tie points to profile III. That is the strategy we adopted. At profile III (across-flow) the sequence is centered under the channel and as reviewer 2 points out, beyond the limits of the channel. But that is to be expected for a fan shaped deposit. We have now strengthened this argument with sedimentary sequence extent at the seabed of profile II.
- The ocean cavity with a steeply descending seabed and thus allowing for a large fan.
- The subglacial channelized flow upstream of the basal channel where these deposits
- The stable grounding line.

Reviewer 2 gives us an additional piece of evidence and a suggestion as to why the sedimentation sequence might indeed be a grounding line fan for which we thank him:
- The sedimentary sequence with chaotic reflections does show seaward dipping reflections.
- The seabed of the sedimentary sequence with chaotic reflections dips with 1.1d values that are found on large glaciomarine fans (Lajeunesse 2002, Dowdeswell 2015)

We adopted this in the text and adjusted figure 2d, 3d and a new Figure 4 accordingly.

Specific comments:

1. East or West Antarctica?
We adjusted the title with Antarctica.

2. Evidence for a grounding line fan:
Our arguments are listed in the intro.
The remark "fairly flat" page 10, line 15 should have said "if the base were fairly flat". It's an example under what conditions the seabed shows up as a mirrored version of the base of the ice shelf at a time migrated profile.

3. Dimensions of the fan and seabed topography:
Rev 2: "The volume of sediment is huge, 5.6 km3."
If I do the math and recalling that the volume is probably closest represented by half the volume of an elliptic cone (and not a rectangle, fan-shaped) with a length of 6.75 km and two radii of 1.6 km and 0.2 km, I come to $1.1 km^3$. This number is still a coarse estimate as a cone probably overestimates the lateral extend of the fan downstream. If I read Dowdeswell 2015 this is not unusual. Examples of these large fans are given by Lajeunesse 2002 at Hudson Bay. These fans are remnants of the Laurentide Ice Sheet that, in volume, comes close to Antarctica. SFG is a huge ice stream and its descending seabed is similar to the setting Lajeunesse describes.

Summarizing: I don't see the size of the fan as unusually large.

"Why do we not see any topographic evidence in profile III?"
This is not necessarily the case, the seabed may topography on the eastern side but this may also be caused by the topography of the basal channel (see adjusted figure 3d). Figure 4a (profile II) clearly shows the sequence is deposited on top of a different formation at a topographic high in the seabed.

4. "What is it specifically about the reflections (e.g. thickness, scale, morphology etc.) that provides diagnostic evidence of deposition by meltwater?"

Facies:
Originally, we named the sequence "disturbed reflections with some stratification". Your comment here was:
- "**How can a material be transparent yet stratified and disturbed?**"
And now is:
- "Based on figure 2d, to me it looks like the reflections, whilst sloping, curved, **disturbed etc. actually have some lateral continuity and stratification,** so their description as 'chaotic' may be a misnomer."

Having checked classes of seismic facies of sedimentation several times now, I did not come across the term "disturbed" and as such I did not adopt this term.
If I were to more accurately and meaningful describe this sedimentation sequence I will opt for: "The sequence is chaotic to weakly stratified. Reflections are mostly curved upward, discontinuous (a lateral extend of 100 to 600 m which to my mind is discontinuous) and occasionally dipping in a seaward direction. The amplitudes are high and a show little signal loss with increasing depth, however boundaries (lower and lateral) are transitional. We'd like refer to this sequence as the **"sedimentary sequence with chaotic reflections"**.

That this is a description of the along-flow profiles is and was obvious as I describe the seismic facies per line. Subchapter (profile I) where I describe it, I originally named along-flow profile I. The direction of the facies description is given per profile and should thus be obvious.

We added the seabed of profile II (Fig 4a) where the reflections are more continuous, stratified and are less chaotic. But as it is the same sequence, we refer to it as **"sedimentary sequence with chaotic reflections"**.

We have explained why we do not describe the seismic facies description of profile III

Why this facies description strengthens the argument for a grounding line fan we argue at the intro

Extent:
Plotting the shots with the sedimentary sequence with chaotic reflections of profiles I, II (withdrawn before but adjusted and resubmitted now) and III and extrapolating these outer boundaries provides "the smoking gun" as to why these deposits most likely come from the

subglacial drainage channel and are fan shaped (Figure 4b). Towards the grounding line the boundaries converge to the subglacial drainage channel at the grounding line (radar profile 6). The point source entrance of the fan derived through extrapolation lies 750 m exactly downstream of the subglacial drainage channel. This small mismatch can have several reasons. A linear extrapolation may be too simple. It can also be that, as with profile I, the seabed does not

5. Agreed
6. Agreed
7. Agreed

8. Figure 2 and 3 are annotated the same way, the seismic profile followed by a schematic interpretation. I have amplified the visibility of the seabed and added a zoom of the seabed. As mentioned, the topography and morphology are influenced by the topography of the ice shelf base so it is a complicated figure. The key message of figure 3d is the lateral extend of the sedimentary sequence only being centered under the channel. The sequence is absent at the eastern and western end. To me that is a strong indicator the subglacial drainage channel is the source.

   Rev 2: "The sequence manifests far beyond the limits of the channel".
   It does, roughly 1.5 km in each direction but not to the ends of profile III. That makes the subglacial channel causing the basal channel a likely source of deposition (see added figure 4b).

9. "Figure 3 – there are very significant differences between 3c and the other two seismic profiles in this figure".

   This is and was clearly explained in the text (was page 13 lines 7-10, now lines 19-22): no the ice-water interface of profile III is not very different than profile IV. The source is different and produces a ghost making the source wavelet longer.

Best regards,

Coen Hofstede